

# Porting the Meso-NH atmospheric model on different GPU architectures for the next generation of supercomputers (version MESONH-v55-OpenACC)

Juan Escobar, Philippe Wautelet, Joris Pianezze, Florian Pantillon, Thibaut Dauhut, Christelle Barthe, and Jean-Pierre Chaboureau

LAERO, Université de Toulouse, CNRS, UT3, IRD, Toulouse, France

**Correspondence:** Jean-Pierre Chaboureau (jean-pierre.chaboureau@univ-tlse3.fr)

**Abstract.** The advent of heterogeneous supercomputers with multi-core central processing units (CPUs) and graphics processing units (GPUs) requires geoscientific codes to be adapted to these new architectures. Here we describe the porting of the Meso-NH version 5.5 community weather research code to GPUs named MESONH-v55-OpenACC, with guaranteed bit reproducibility thanks to its own `MPPDB_CHECK` library. This porting includes the use of OpenACC directives, specific memory
management, communications optimization, development of a geometric multigrid solver and creation of an in-house preprocessor. Performance on AMD MI250X GPU Adastra platform shows up to 6.0× speedup (4.6x on NVIDIA A100 Leonardo platform), and achieves a gain of a factor 2.3 in energy efficiency compared to AMD Genoa CPU Adastra platform, using the same configuration with 64 nodes. The code is even 17.8 faster by halving the precision and quadrupling the nodes with a gain in energy efficiency of a factor 1.3. First scientific simulations of three representative storms using 128 GPUs nodes of Adastra
show successful cascade of scales for horizontal grid spacing down to 100 m and grid size up to 2.1 billion points. For one of these storms, Meso-NH is also successfully coupled to the WAVEWATCH III wave model via the OASIS3-MCT coupler without any extra computational cost. This GPU porting paves the way for Meso-NH to be used on future European exascale machines.

## 1 Introduction

Numerical simulation of the atmosphere plays a crucial role in understanding and anticipating extreme weather phenomena. The skill of numerical weather prediction (NWP) has continuously improved over the last few decades, thanks to a steady accumulation of scientific knowledge and technological advances (Bauer et al., 2015). Increased computing power has enabled numerical simulation to represent even greater complexity on a wider range of scales. Current operational NWP codes typically achieve horizontal resolutions of $\mathcal{O}(10\,\mathrm{km})$ on a global scale and $\mathcal{O}(1\,\mathrm{km})$ on a regional scale, which represent deep convection
in parameterized and explicit ways, respectively. Global storm-resolving models used in research can simulate the atmosphere globally with convection-permitting resolutions of $\mathcal{O}(1\,\mathrm{km})$ to seamlessly represent scales from local storms to planetary waves (e.g., Tomita et al., 2005; Fuhrer et al., 2018; Stevens et al., 2019; Giorgetta et al., 2022; Donahue et al., 2024).



The Meso-NH community weather research code (Lac et al., 2018) takes advantage of an efficient parallelization and from increases in computing power to run large-eddy simulations (LES; horizontal resolution of $\mathcal{O}(10$–$100$ m)) on very large computational grids of the order of a billion grid points. These so-called giga-LES (Khairoutdinov et al., 2009) resolve most of the turbulent kinetic energy over the full extent of weather systems ($\mathcal{O}(100$ km)). Such giga-LES runs with Meso-NH make it possible to show, for example, the effect of small-scale surface heterogeneities and buildings on the radiation fog over an airport (Bergot et al., 2015), turbulent mixing leading to stratospheric hydration by overshooting convection (Dauhut et al., 2018), the multiscale modeling of coupled radiative and heat transfer in complex urban geometry (Villefranque et al., 2022), and the downward transport of strong winds by roll vortices in a Mediterranean windstorm (Lfarh et al., 2023). Designed for studies of scale interactions between processes, these giga-LESs provide a unique research base for statistical or climate studies in a scientific community beyond that of Meso-NH.

To benefit from technological advances, numerical codes of the atmosphere need to be adapted to current and future supercomputers based on hybrid architectures with both central processing units (CPUs) and graphical processing units (GPUs). For instance, the most powerful system today is Frontier in the USA, a nearly 2 exaFLOPS machine (i.e., exceeding $2 \times 10^{18}$ floating point operations per second) based on AMD MI250X GPUs. In Europe, while the most powerful systems are below the exascale level (i.e., below 1 exaFLOPS), they are also built with GPUs, as for Leonardo in Italy with NVIDIA A100 GPUs (0.3 exaFLOPS) and for Adastra in France with AMD MI250X GPUs (0.06 exaFLOPS). Running a model like Meso-NH on these new supercomputers is a big challenge because the code must be ported to GPU. An abundant literature documents the efforts made by the various modeling communities. For example, Fuhrer et al. (2018) have completely rewritten the dynamical core of the COSMO model using a domain-specific language. Giorgetta et al. (2022) have ported the ICON atmosphere model to GPUs by introducing OpenACC directives. After 5 years of model development, Donahue et al. (2024) have fully rewritten the Energy Exascale Earth System Model atmosphere model in C++ using the Kokkos library.

Here, we describe the porting of Meso-NH version 5.5 to GPU named MESONH-v55-OpenACC and illustrate applications with giga-LES runs in the framework of a grand challenge on the Adastra supercomputer, ranked 10th in the June 2022 TOP500 (TOP500.org, 2022b) and 3rd in the November 2022 GREEN500 (TOP500.org, 2022a). We choose the directive-based method using OpenACC for the porting. First, this method of manually adding GPU directives has less impact on the code structure, thus limiting the human effort involved in porting the code. Second, it preserves the readability of the original code. This point is of particular importance for a research code developed and used by many different people and for many different applications in the fields of meteorology, air quality and surface coupling. Third, this method allows the same source code to run on either CPU or GPU. This is another crucial point for Meso-NH, which runs on different computing architectures, from personal computers to supercomputers in regional, national and international facilities. Fourth, this method enables step-by-step verification of the addition of GPU modifications, and thus allows detection of any mistake in implementation or compiler bug. This is of increased importance for the quality of Meso-NH, which is bit-reproducible on CPU. Indeed, Meso-NH provides the same results whatever the number of CPUs, which is verified using our own `MPPDB_CHECK` library. This bit-reproducibility of Meso-NH has been extended here to the OpenACC GPU version, providing bit-reproducibility from CPU execution to GPU execution, and even-more so to multi-GPU execution. To our knowledge, Meso-NH is the only atmospheric (or oceanic) model



providing this outstanding capability, which guarantees bug-free implementation for massively parallel executions on CPU or GPU supercomputers.

The remainder of the paper is organized as follows. Section 2 details the methodology with a brief overview of Meso-NH (Sect. 2.1), the inclusion of OpenACC directives (Sect. 2.2), checking bit-reproducibility between CPU and GPU (Sect. 2.3), memory management replacing the use of automatic or allocatable arrays to reduce overhead (Sect. 2.4), the optimization of communications with a GPU-aware MPI library (Sect. 2.5), the development of a geometric multigrid solver (Sect. 2.6) and the creation of an in-house preprocessor to facilitate the support and optimization of Meso-NH on different architectures (Sect. 2.7).

Section 3 presents performance achieved on a single node and the scaling up to multiple nodes across multiple platforms. Section 4 describes several large-grid weather applications running on the Adastra supercomputer. Section 5 concludes the paper.

## 2   Methodology

### 2.1   The Meso-NH community weather research code

Meso-NH is a community weather research code (Lac et al., 2018), initially developed by the Centre National de Recherches Météorologiques (CNRS and Météo-France) and the Laboratoire d'Aérologie (LAERO; UT3 and CNRS). It is a grid-point limited-area model based on a non-hydrostatic system of equations to handle a wide range of atmospheric phenomena, from synoptic to turbulent scales. The code includes fourth-order centered and odd-order WENO advection schemes for momentum and monotonic advection schemes for scalar transport (Lunet et al., 2017). It has a complete set of physical parameterizations,

including clouds, turbulence and radiation. Meso-NH is coupled with the SURFEX surface model (Masson et al., 2013) and can be coupled with any ocean or wave models that includes OASIS code instructions (Voldoire et al., 2017). Since Meso-NH is based on an anelastic continuity equation (Lafore et al., 1998), an elliptic equation must be solved with great precision to determine the pressure perturbation. The current pressure solver consists on a conjugate-residual algorithm accelerated by a flat fast Fourier transform (FFT) preconditioner.

Meso-NH is written mainly in Fortran 95 with the use of some more recent functionalities from Fortran 2003 and 2008. It is fully vectorized, i.e., it uses array syntax with almost no loops. Since 1999, most of the code is parallel (Jabouille et al., 1999). The 3-D domain is split into horizontal subdomains in the $x$ and $y$ directions. Each subdomain is then assigned to one process on the parallel computer, and an interface package based on the standard MPI (Message Passing Interface) library ensures communications between the processes. In 2011, parallel capability was extended to petaFLOPS computers (i.e., exceeding

$10^{15}$ floating point operations per second) by allowing the grid to be sliced during input/output (I/O) into horizontal planes and parallelizing the FFT preconditioner vertically and horizontally (Pantillon et al., 2011).

   The standard Meso-NH benchmark is "Hector The Convector", a case of very deep convection that occurs over the Tiwi Islands, North of Darwin, Australia (Dauhut et al., 2015). This test case is easy to run on any supercomputer, because it is initialized in temperature, humidity and wind with a single sounding and applies open boundary conditions. In this paper, the

weather applications use initial and lateral boundary conditions provided either by the operational analyses of the European



Centre for Medium-Range Weather Forecasts (ECMWF) Integrated Forecasting System (IFS), or the Météo-France Applications de la Recherche à l'Opérationnel à Méso-Echelle (AROME) code. The benchmark and weather application runs on Adastra AMD MI250X GPUs nodes include the most commonly used transport schemes and physical parameterizations in Meso-NH. Momentum variables are advected with a centred fourth-order scheme while scalar variables are advected with

the Piecewise Parabolic Method (PPM) advection scheme (Colella and Woodward, 1984). The physical parameterizations are a 1.5-order closure scheme for turbulence (Cuxart et al., 2000) and a one-moment bulk microphysics scheme named ICE3 (Pinty and Jabouille, 1998) including five water species (cloud droplets, raindrops, pristine ice crystals, snow / aggregates, and graupel). The simulations also involve a radiation scheme, usually called every 30 or 60 time steps. The latter coming from the ECMWF, no attempt of porting to GPU has been done.

Porting the complete code to GPU is a huge task, as Meso-NH contains several thousand source files totaling about a million lines of code. However, the Pareto principle holds for Meso-NH, so that 90% of the computation time comes from 10% of the code. Thus, the porting work mainly concerns the most computationally intensive parts of Meso-NH, that is advection, turbulence, cloud microphysics and the pressure solver. This porting work is the result of a development initiated in the early 2010s on a NVIDIA GPU using OpenACC directives with the PGI compiler (since then acquired by NVIDIA).

More recently, from late 2021, it has continued with the start of implementation on AMD GPUs using the Cray compiler. In the following, the changes made to port Meso-NH to NVIDIA and AMD GPUs are detailed. The overall impact that leads to the MESONH-v55-OpenACC version is summarized in Table 1. The left column shows the changes made in Meso-NH before the port to AMD GPUs (i.e., for the initial port to NVIDIA GPUs), and the right column the state after it (i.e., for the port to both GPU types). This results in the inclusion of thousands of OpenACC directives. Memory management routines

are frequently used, among other things, to reduce the performance impact of allocations and deallocations. Calls to bit-reproducible mathematical functions appear wherever they are necessary. Moreover, loops in array syntax are replaced, in some cases, by `do concurrent` constructs. For AMD GPUs, an in-house preprocessor was developed, leading to its use in more than one thousand occurrences, and a reduction in the number of `do concurrent` loops.

## 2.2   Inclusion of OpenACC directives

The OpenACC paradigm offers a promising approach for Meso-NH. The developer only has to add directives (seen as comments) in the code to port it to GPUs. Moreover, since the Meso-NH code is mostly written in array syntax, supported by OpenACC, this syntax, free of loops, is well suited to auto-parallelization and auto-vectorization. To limit the impact on the source code and facilitate the porting work, the use of the auto-parallelization `kernels` directives is preferred. The code largely resembles the one shown in Listing 1.

---

**Listing 1** OpenACC kernels directives with array syntax

---

```
!$acc kernels
A(:,:,:) = B(:,:,:) + C(:,:,:)
D(:,:,:) = E(:,:,:) + F(:,:,:)
!$acc end kernels
```

---



**Table 1.** Number of modified Meso-NH lines for the initial GPU porting on NVIDIA GPUs and after the porting on AMD GPUs. Other changes (multigrid solver, memory manager (except calls to it), algorithmic modifications, code transformations, and optimizations) are not accounted for.

| OpenACC compute directives | NVIDIA GPU | AMD GPU |
|---|---|---|
| `!$acc kernels` | 1917 | 1980 |
| `!$acc loop (collapse) independent` | 533 | 151 |
| `!$acc ... async` | 220 | 256 |
| `!$acc ... wait` | 78 | 79 |
| `!$acc loop seq` | 18 | 43 |
| `!$acc atomic` | 13 | 13 |
| OpenACC data directives | Number of lines | |
| `!$acc ... data` | 463 | 544 |
| `!$acc host_data use_device` | 16 | 80 |
| Fortran | Number of lines | |
| Memory management routine calls | 637 | 1776 |
| Bit-reproducibility function calls | 619 | 665 |
| `do concurrent` | 293 | 13 |
| In-house preprocessor directives | Number of lines | |
| `!$mnh_do_concurrent` | | 502 |
| `!$mnh_expand_array` | | 111 |
| `!$mnh_expand_where` | | 19 |
| `!$acc ... present_cr` | | 694 |
| `!$mnh_define`/`!$mnh_undef` | | 45 |

The drawback of the OpenACC paradigm is the data location. Since the memory of the CPU and the GPU are usually separate, the developer must carefully manage data location and transfers between them by adding appropriate OpenACC directives. It is easy to make mistakes and to introduce bugs, or have poor performance if unnecessary transfers are coded. Adding OpenACC directives is not always sufficient because for optimization or to work around compiler bugs some loops need to be explicitly written, which results in losing the compact array syntax. Array-returning functions are not parallelized on

GPU (at least in our early developments). They are replaced by subroutines and the use of temporary arrays to store intermediate results (Listing 2).



---

**Listing 2** Example of replacing an array-returning function by a subroutine using temporary arrays to store intermediate results. Note that the original version is preserved and compiled if the preprocessor key `MNH_OPENACC` is not set.

---

```
#ifndef MNH_OPENACC
PRUS(:,:,:)  =  PRUS(:,:,:)       &
        −  DXM(MXF(PRUCT(:,:,:)))  &
                    *  ZMEANX(:,:,:)  )
#else
call  MXF_DEVICE(PRUCT,ZTEMP1)
!$acc kernels
ZTEMP2(:,:,:)  =  ZTEMP1(:,:,:)  &
                    *  ZMEANX(:,:,:)
!$acc end kernels
call  DXM_DEVICE(ZTEMP2,ZTEMP3)
!$acc kernels
PRUS(:,:,:)  =  PRUS(:,:,:)       &
                    −  ZTEMP3(:,:,:)
!$acc end kernels
#endif
```

---

## 2.3 Verification of bit-reproducibility between CPUs and GPUs

The original CPU-only Meso-NH code, running in parallel on CPU clusters using the MPI library, is already bit-reproducible when the number of MPI tasks varies. This guarantees that no parallelization bugs have been introduced in the CPU coding.
This is achieved using our own `MPPDB_CHECK` library (Fig. 1). The principle is to run two similar simulations concurrently. The primary simulation launches the replica one with a call to `mpi_comm_spawn`. At certain points, the `MPPDB_CHECK` library checks on the fly that array values are exactly the same down to the bit. These two executions can have any number of MPI processes.

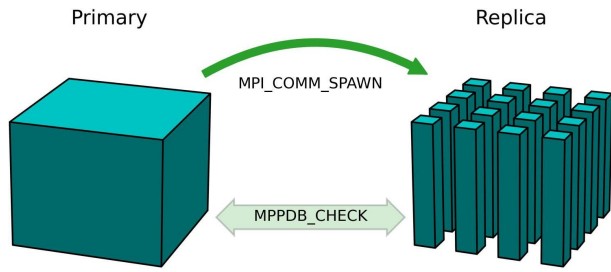

**Figure 1.** Schematic of bit-reproducible verification between primary and replica simulations using the `MPPDB_CHECK` library.

The internal `MPPDB_CHECK` library has been ported to GPU, with the ability to compare values stored in CPU or GPU
memory. For NVIDIA GPUs, the Meso-NH code is compiled with the NVIDIA compiler by setting the `-acc=host,gpu` flag. In this way, the same executable can run on both CPUs and GPUs. The primary simulation runs on CPUs with `ACC_DEVICE_TYPE=HOST` while the replica runs on GPUs with `ACC_DEVICE_TYPE=NVIDIA`. Validation of code changes in Meso-NH for GPU porting is a step-by-step process. To validate the port, the results must be exactly the same





on both sides. For AMD GPUs, it is not possible to compile a single executable that can run indifferently on CPU or GPU with
the Cray compiler. Therefore, two different executables are generated, one for CPU only and one that supports GPU offloading.
To ensure bit reproducibility, options `"-Kieee -Mnofma -gpu=nofm"` are passed to the NVIDIA compiler to enforce
compliance with the IEEE 754 standard (IEEE, 2019) and to disable FMA (Fused Multiply-Add) instructions. For the Cray
compiler, options `"-Ofp0 -hnofma"` are provided.

Some operations are optional ("recommended" but not "required") in the IEEE 754 standard, such as sqrt, exp, log, ....
Several rounding modes are also possible. The sqrt standard implementation is available with NVIDIA and Cray compilers
and architectures (if certain compiler options mentioned above are enabled) and gives exactly the same result on both CPU
and GPU. This is not true for other mathematical operations provided by the compiler-linked mathematical libraries that are
not bit-reproducible between CPU and GPU. It is therefore necessary to replace the intrinsic functions log, pow, sin, cos,
atan and atan2. They have all been replaced with equivalent operations based on reproducible standard operations. This
is achieved using the bitrep library (Spiros, 2014) written in C++. The performance of these functions tends to be lower,
but in practice the impact on runtime is very limited (less than a few percent). As they are only used to check the correct
implementation of the port, their use could be disabled to restore performance. However, the bit-reproducible version is kept
even for the run tests presented in this paper.

The bitrep C++ functions have been ported to GPU (Listing 3). This is done by adding OpenACC directives for the NVIDIA
compiler. Since the Cray C++ compiler does not support OpenACC directives natively, OpenMP directives are used as an
alternative. This implementation seamlessly integrates the OpenACC and OpenMP paradigms, allowing for efficient execution
on either NVIDIA or Cray environments. This hybrid approach enables us to leverage the strengths of each programming
model while ensuring compatibility with the targeted hardware.

---

**Listing 3** Extract of the porting of the log function of the bitrep library. OpenMP and OpenACC directives are used.

```cpp
namespace bitrep {
#ifdef MNH_BITREP_OMP
#pragma omp declare target
#else
#pragma acc routine seq
#endif
double log(double x)
{...}
}
// Implement C interface
extern "C"
{
#ifdef MNH_BITREP_OMP
#pragma omp declare target
#else
#pragma acc routine seq
#endif
double br_log(double x) \
    {return bitrep::log(x);}
}
```

---





To call these C/C++ functions from Meso-NH, a C-Fortran interface has been written (Listing 4). All the new bit-reproducible

math functions are prefixed with `BR_`. An example of using the logarithm function is shown in Listing 5.

---

**Listing 4** Extract of the C-Fortran interface for the `br_log` function.

```
ELEMENTAL FUNCTION BR_LOG(PVAL)
!$acc routine seq
REAL, INTENT(IN) :: PVAL
REAL           :: BR_LOG
INTERFACE
   PURE FUNCTION BR_LOG_C(PIN) BIND(C,NAME="br_log")
!$acc routine seq
      IMPORT C_DOUBLE
      REAL(KIND=C_DOUBLE)               :: BR_LOG_C
      REAL(KIND=C_DOUBLE),VALUE,INTENT(IN) :: PIN
   END FUNCTION
END INTERFACE
BR_LOG = BR_LOG_C(REAL(PVAL,KIND=C_DOUBLE))
END FUNCTION
```

---

**Listing 5** An example of using the logarithm function. The `MNH_BITREP` preprocessor key is used to choose between the original and bit-reproducible versions.

```
!$acc kernels
DO CONCURRENT( JI = 1 : JIU )
#ifndef MNH_BITREP
   ZZW(JI) = LOG(PT(JI))
#else
   ZZW(JI) = BR_LOG(PT(JI))
#endif
END DO
!$acc end kernels
```

---

### 2.4 Memory management

In Meso-NH, memory management for arrays consists mainly of automatic or allocatable arrays. Due to various compiler bugs and the very poor performance of memory allocation on GPUs, this management has been redesigned for use on the GPU. A relatively simple approach is adopted. In the initialization phase, a large 1D array is allocated once in both CPU and

GPU memory spaces. This operation is done for each main intrinsic data type (real, integer and logical). These arrays are used as a memory pool in which the different variable arrays are stored. Therefore, no more array allocation is needed during calculations. The size of these arrays must be carefully chosen to be large enough to contain all the required data, but not so large as to waste memory. Their dimensions can be selected using a parameter read from a standard Meso-NH namelist at simulation start, without the need to recompile the model.

When necessary, usually at the beginning of a subroutine (Listing 6), and instead of relying on automatic or dynamic allocation, allocations are replaced by calls that provide pointers to unique portions of the memory pool with `MNH_MEM_GET`. To keep memory management as simple as possible, before a series of "allocations", the positions of the pointers within the pools



are stored using `MNH_MEM_POSITION_PIN`. When the memory is no longer needed (usually at the end of the subroutine), the pointers are reset to these values with `MNH_MEM_RELEASE`. This is a LIFO (Last-In-First-Out) approach, which is ensured

by providing the same dummy argument before the block of "allocations" and the "release" of the corresponding data. In this way, no deallocation or clean-up is required and no memory fragmentation is possible.

---

**Listing 6** Example of code structure for memory management.

```fortran
! Variable declaration
#ifndef MNH_OPENACC
INTEGER, DIMENSION(:),      ALLOCATABLE :: I1
LOGICAL, DIMENSION(:,:,:),  ALLOCATABLE :: GMICRO
REAL,    DIMENSION(:,:,:),  ALLOCATABLE :: ZRR
#else
INTEGER, DIMENSION(:),      POINTER, CONTIGUOUS :: I1
LOGICAL, DIMENSION(:,:,:),  POINTER, CONTIGUOUS :: GMICRO
REAL,    DIMENSION(:,:,:),  POINTER, CONTIGUOUS :: ZRR
#endif

! Data allocation (beginning of the subroutine)
#ifndef MNH_OPENACC
ALLOCATE( I1     (IIJKU)         )
ALLOCATE( GMICRO(IIU,IJU,IKU) )
ALLOCATE( ZRR    (IIU,IJU,IKU) )
#else
!Pin positions in the pools of MNH memory
CALL MNH_MEM_POSITION_PIN( 'RAIN_ICE' )
...
CALL MNH_MEM_GET( I1,        IIJKU  )
CALL MNH_MEM_GET( GMICRO,   IIU,IJU,IKU )
CALL MNH_MEM_GET( ZRR,       IIU,IJU,IKU )
#endif

! Data release (end of the subroutine)
#ifndef MNH_OPENACC
DEALLOCATE( I1, GMICRO, ZRR )
#else
!Release all memory
CALL MNH_MEM_RELEASE( 'RAIN_ICE' )
#endif
```

---

To reduce the number of memory pools to just one per intrinsic data type, pointer bounds remapping (a Fortran 2003 functionality) is used to map multidimensional arrays onto 1D pools. A possible optimization is to align the memory addresses of the beginning of each returned pointer to GPU memory segments, but tests on NVIDIA and AMD GPUs show no performance

improvement. The use of pointers instead of automatic or allocated arrays may prevent the compiler to do some optimizations because it must assume that aliasing is possible between different pointers. This behavior has a significant impact on the performance with the Cray compiler used for the porting to AMD GPUs. To restore good performance, it is necessary to pass the `"-halias=none"` option to this compiler.





## 2.5 Communications with a GPU-aware MPI library

When run on a supercomputer with multiple CPUs and GPUs, Meso-NH uses the MPI library and domain decomposition with additional halo points at grid boundaries in horizontal directions to maintain computational consistency. For halo exchange communications on the GPU, the data is already in GPU memory, as it is computed there. It is therefore more efficient to transfer data directly between GPUs, or within a given GPU if the processes run on the same GPU. Otherwise, the data is copied to CPU memory, transferred between CPUs, and finally copied to the memory of the target GPU. On some architectures,

such as Adastra, network cards are physically connected to GPUs rather than CPUs, and the available MPI library is GPU-aware, allowing direct transfer between GPUs. On other architectures, even if the network is not directly connected to the GPUs, direct transfers between processes running on GPUs inside a machine node are possible if the MPI library supports them. For OpenACC coding, this is implemented by the directive `!$acc host_data use_device`. An extract from the `GET_HALO_START_D` and `GET_HALO_STOP_D` subroutines is shown in Listing 7. It details how this GPU-aware capability

is combined with MPI non-blocking communications and OpenACC asynchronous kernels.

In the `GET_HALO_START_D` routine, halo communication is initiated. First, a non-blocking receive (`MPI_IRECV`) is posted in advance for the output halo buffer (`PZSOUTH_OUT`), indicating that the data is stored in GPU memory (`!$acc host_data use_device(PZSOUTH_OUT)` directive). Then, in an asynchronous OpenACC kernel (`!$acc kernels async(IS_NORTH)`), the input halo buffer (`PZNORTH_IN`) is filled from the boundary of the field

`PSRC`. The same operation (not shown) is done for the south, east and west boundaries. To ensure completion of the four kernels filling the north, south, east and west buffers, an OpenACC synchronization is performed with `!$acc wait`. At the end of the routine, the input halo buffer `PZNORTH_IN` is sent with a non-blocking `MPI_ISEND` call, encapsulated by the OpenACC directive indicating again that the data is on the GPU with `!$acc host_data use_device(PZNORTH_IN)`. In the `GET_HALO_STOP_D` routine, halo communication is finalized. First, the completion of the previous non-blocking MPI

communication is ensured with a `MPI_WAITALL` call. Next, the output halo buffer `PZSOUTH_OUT` is copied to the southern boundary of the field `PSRC` in an asynchronous OpenACC kernel (`!$acc kernels async(IS_SOUTH)`). The same operation is repeated for the three other boundaries in three other asynchronous kernels (not shown). Finally, the subroutine ends by waiting for the completion of these kernels (`!$acc wait`). Between calls to the `GET_HALO_START_D` and `GET_HALO_STOP_D` routines, operations not dependent on the field involved in the halo exchange can be interspersed. This

allows calculations to overlap with communications.

## 2.6 Development of a multigrid pressure solver

A critical point for GPU porting is the pressure solver needed for the elliptic equation inversion. In the original version of Meso-NH, this solver consists of an FFT pre-conditioner associated with the conjugate-residual algorithm. The FFT algorithm requires all-to-all communications between MPI processes and therefore between GPUs when several GPUs are used. These

data transfers are very bandwidth-intensive, and their cost increases rapidly with the number of GPUs. As the local FFT



---

**Listing 7** Extract of MPI GPU-aware usage for north-south halo exchange (west-east removed) with MPI non-blocking communications and OpenACC asynchronous kernels

```fortran
SUBROUTINE GET_HALO_START_D(PSRC,...)
  ...
  IF (.NOT.GSOUTH) THEN
     !$acc host_data use_device(PZSOUTH_OUT)
     CALL MPI_IRECV(PZSOUTH_OUT, SIZE(PZSOUTH_OUT), MNHREAL_MPI, NP_SOUTH-1, &
                    1000+IS_NORTH, NMNH_COMM_WORLD, KREQ(NB_REQ), IERR)
     !$acc end host_data
  ENDIF
  ...
  IF (.NOT.GNORTH) THEN
     !$acc kernels async(IS_NORTH)
     PZNORTH_IN(KIIB:KIIE,KIJE-KIHALO_1:KIJE,:) = PSRC(KIIB:KIIE,KIJE-KIHALO_1:KIJE,:)
     !$acc end kernels
  ENDIF
  ...
  !$acc wait
  ...
  IF (.NOT.GNORTH) THEN
     !$acc host_data use_device(PZNORTH_IN)
     CALL MPI_ISEND(PZNORTH_IN, SIZE(PZNORTH_IN), MNHREAL_MPI, NP_NORTH-1, &
                    1000+IS_NORTH, NMNH_COMM_WORLD, KREQ(NB_REQ),IERR)
     !$acc end host_data
  ENDIF
END SUBROUTINE GET_HALO_START_D

SUBROUTINE GET_HALO_STOP_D(PSRC,...)
  ...
  CALL MPI_WAITALL(NB_REQ,KREQ,MPI_STATUSES_IGNORE,IERR)
  ...
  IF (.NOT.GSOUTH) THEN
     !$acc kernels async(IS_SOUTH)
     PSRC(KIIB:KIIE,1:KIJB-1,:) = PZSOUTH_OUT(KIIB:KIIE,1:KIJB-1,:)
     !$acc end kernels
  ENDIF
  ...
  !$acc wait
END SUBROUTINE GET_HALO_STOP_D
```

---

calculations run faster on GPU than on CPU, the fraction of time consumed by these communications is not negligible and can become very high, especially when multiple GPUs distributed across multiple nodes are used.

As the FFT solver tends to be inefficient as the number of GPUs increases, a more efficient algorithm is required. The most promising alternative for solving this type of elliptic equation is the use of a geometric multigrid solver for regular

structured grids (Fig. 2). To our knowledge, there is no Fortran numerical library ported with OpenACC that provides such tool. Consequently, the Fortran version of the TensorProductMultiGrid solver developed by Müller and Scheichl (2014) was selected. This multigrid (MG) solver developed for the UK Met Office is well-suited to NWP models with a highly vertically stretched grid. The original code is a standalone benchmark version, and many modifications have been made to integrate it into Meso-NH.





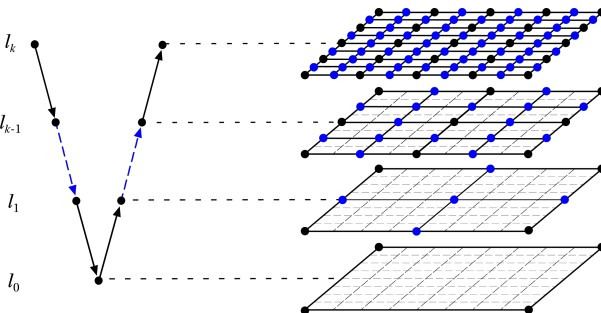

**Figure 2.** Schematic of the V-cycle of the geometric multigrid pressure solver on the horizontal grid. Starting from the finest grid at level $k$ with $2^k$ cells in the $x$ and $y$ directions, the restriction phase is performed on the coarse grid at level $k-1$, which is obtained by halving the number of cells in each direction. The process proceeds from top to bottom until the grid contains only one cell. Then, in the prolongation phase, the grids are refined in the reverse process upward, multiplying the grid cells in each direction by 2 (blue points figuring edges added/removed in this process).

The MG code was first adapted for the CPU version of Meso-NH. On the Meso-NH part, interface routines were introduced. On the MG part, the tensor-product coefficients were fitted to meet the needs of Meso-NH; new Neumann boundary conditions specific to Meso-NH were added. For debugging and bit-reproducibility purposes, calls to our `MPPDB_CHECK` library were incorporated into the MG code. This revealed a bug in the original parallel code (a missing MPI halo exchange). After fixing, a fully bit-reproducible on multiple CPUs version of Meso-NH with this new solver was obtained. Extensive performance and
scalability tests were performed to set optimal parameters for the solver, such as the iterative method, the convergence rate, the smoother, the number of grid levels, the restriction and prolongation phases and the coarsest-grid solver.

Due to the characteristics and the original implementation of the MG solver, its use imposes some constraints. First, the grid size must be of the form $2^N$ for both horizontal axes, as shown in Fig. 2. At each level of coarsening, from $k$ to $k-1$, the number of cells is halved in each direction until the last level which contains only one cell. Second, the grid size on the
vertical axis is free, since no downsizing is done in this direction due to strong stretching that makes it unfit. In this direction, since the matrix to be inverted has a tridiagonal form, a Thomas algorithm (simplified form of Gaussian elimination) is used instead. Finally, as the original solver does not allow for load imbalance, the number of MPI processes must be $2^{(2\times P)}$ because the size of the solver grids must be halved in each direction at each coarsening. This limitation has been relaxed somewhat. The number of MPI processes can be of the form $2^P$, but the MG must use one less grid level. In that case, the last coarse level
can have $2\times2$ or $2\times1$ cells, and two iterations are needed in this last level.

To port this multigrid pressure solver on GPUs, the MG code has been refactored to change the original layout of the arrays from KJI to IJK indexing (as in Meso-NH) to optimize memory accesses on GPUs, and OpenACC `kernels` directives have been added. The same recipes as for Meso-NH have been applied: memory management with a preallocated memory pool, replacement of array syntax with `do concurrent` constructs where necessary, and MPI communications with GPU-aware
OpenACC directives to avoid unnecessary data transfers between CPUs and GPUs. A new optimization is also introduced



specifically for the MG solver for hybrid executions on CPUs and GPUs: a new parameter, configurable at runtime, is added to select the coarse grid level at which the MG solver switches calculations from GPUs to CPUs. This parameter is useful when the local sub-grid is too small to provide performance gains on GPUs. Bit-reproducibility is again ensured between CPU and GPU executions.

Despite the limitations of the FFT pressure solver, it is also ported on GPUs for comparison with the new multigrid solver. The FFT solver has also the advantage of fewer restrictions on grid dimensions and the number of MPI processes. Test results (Section 3) show that it remains a good choice for simulations with a limited number of GPUs.

## 2.7 Creation of an in-house preprocessor

Loop performance optimizations for AMD or NVIDIA GPUs are often incompatible and cannot be mixed. To avoid having two
different versions of the Meso-NH code, and to avoid degrading readability by multiplying preprocessing keys (i.e., `#ifdef` keywords), a small preprocessor named `mnh_expand` has been developed. It uses filepp (Miller, 2008), an enhanced programmable preprocessor, compatible and similar to cpp, with powerful user-defined macros, all written in Perl. With this tool, different transformations can be applied on different architectures allowing customized optimizations on different CPUs, GPUs and compilers.

For NVIDIA GPUs, it is necessary to rewrite some loops originally in array syntax as nested loops or with `do concurrent` constructs. This approach works very well and generates parallel and collapsed loops that are optimal for the GPU. For AMD GPUs, the `do concurrent` syntax is not managed efficiently by the Cray compiler, which generates poor parallel uncollapsed loops. In addition, inclusion of the OpenACC directives which collapses and parallelizes loops (`!$acc loop collapse(X) independent` directive) generates compiler errors or serializes kernels. However, this
directive leads to good parallelization if used in conjunction with nested loops instead of a `do concurrent` construct. The exception is when our bit-reproducible mathematical functions are present in the loop. In the latter case, it is necessary to inhibit the transformation of array syntax into nested loops and not to supply the collapse OpenACC directive (Listing 8).

Optimization choices are made via preprocessor keys. For example, the `MNH_EXPAND_OPENACC` key generates loop collapse and independent OpenACC directives and the `MNH_EXPAND_LOOP` transforms the code into nested loops. If the latter
is omitted, the `do concurrent` instructions are added instead. To manage all these different and contradictory situations, three different macros are introduced (Listing 9). At the preprocessing stage, they generate the most efficient form of expression according to the architecture and compiler targeted.

Loops in Fortran array syntax requiring transformation are enclosed by the `!$mnh_expand_array` and `!$mnh_end_expand_array` directives (Listing 10). They allow automatic rewriting of the loop index in array expres-
sions and generate the appropriate nested loops or `do concurrent` constructs depending on the activated keys (Listings 11 and 12). For expressions already transformed into the `do concurrent` form, the macro `!$mnh_do_concurrent` is used. For `where` constructs in array syntax, the `!$mnh_expand_where` directive is implemented.

Another preprocessor directive is necessary to circumvent the frequent erroneous detection of recurrences between loop iterations by the Cray compiler. These recurrences prevent the parallelization of these loops and lead to the generation of



---

**Listing 8** Example of using the `!$mnh_expand_array` macro for the not-bit-reproducible math functions `exp` and `power`. For the Cray compiler, expansion to nested loops and the addition of OpenACC directives are first inhibited around bit-reproducible procedures, then reactivated for the remaining of the code with the `mnh_undef` and `mnh_define` macros.

```
#if defined(MNH_COMPILER_CCE) && defined(MNH_BITREP_OMP)
!$mnh_undef(LOOP)
!$mnh_undef(OPENACC)
#endif
!$mnh_expand_array(JI=IIB:IIE,JJ=IJB:IJE,JK=1:JKU)
    ZCFLU(IIB:IIE,IJB:IJE,:) = ZCFLU(IIB:IIE,IJB:IJE,:) &
      * (1.-Br_exp(-Br_pow(XIBM_LS(IIB:IIE,IJB:IJE,:,2) &
      / Br_pow(XRHODJ(IIB:IIE,IJB:IJE,:)               &
      / XRHODREF(IIB:IIE,IJB:IJE,:),1./3.),2.)))
!$mnh_end_expand_array()
#if defined(MNH_COMPILER_CCE) && defined(MNH_BITREP_OMP)
!$mnh_define(LOOP)
!$mnh_define(OPENACC)
#endif
```

---

**Listing 9** Synthetic example of the three main in-house preprocessing directives.

```
! For array syntax. Indices are replaced in the same order
! (ii is for the 1st dimension, ij for the 2nd and ik for the 3rd)
!$mnh_expand_array(ii=iib:iie:iis, ij=ijb:ije:ijs, ik=ikb:ike:iks)
 a(:,:,:) = b(:,:,:) * c(:,:,:)
!$mnh_end_expand_array(comment)

! For where
!$mnh_expand_where(ii=iib:iie:ip, ij=ijb:ije:ijs, ik=ikb:ike:iks)
  where ( d(:,:,:) < 0.0 )
    d(:,:,:) = 0.0
  end where
!$mnh_end_expand_where(comment)

! For nested loops
!$mnh_do_concurrent(ii=iib:iie:ip, ij=ijb:ije:ijs, ik=ikb:ike:iks)
  a(ii,ij,ik)=b(ii,ij,ik,...)
!$mnh_end_do(comment)
```

---

serial code. The bypass found is to add extra OpenACC `present` clauses with `loop` or `kernels` directives for certain variables, even if they are already in a block where they have been declared as present. Since, in some cases, the NVIDIA compiler reports errors if a variable is declared present twice, different behaviors are necessary for these two compilers. The `present_cr` macro (Listing 10) generates the OpenACC `present` clause only by activating the `MNH_COMPILER_CCE` preprocessor key (set for the Cray compiler).

## 3 Computational performance

The computational performance is estimated for the "Hector The Convector" test case, the standard Meso-NH benchmark. The test case uses advection, turbulence, cloud microphysics, pressure solver and other components. These include radiation, which is called every 900 s only. The vertical grid has 128 levels. The horizontal grid covers a square of 204.8 km side centered





---

**Listing 10** Example of the use of the `!$mnh_expand_array` and `present_cr` macros

```
! dyn_sources.f90
...
!$acc data present(ZWORK1,ZWORK2)
...
!$acc kernels present_cr(ZWORK1,ZWORK2)
  !$mnh_expand_array(JI=1:JIU, JJ=1:JJU, JK=1:JKU)
    ZWORK1(:,:,:) = PCURVX(:,:) / XRADIUS
    ZWORK2(:,:,:) = PCURVY(:,:) / XRADIUS
  !$mnh_end_expand_array()
!$acc end kernels
...
!$acc end data
...
```

---

**Listing 11** Example of transformed sources after preprocessing the `!$mnh_expand_array` and `present_cr` macros for the NVIDIA compiler. Note that the `present_cr` clause has vanished; otherwise, the NVIDIA compiler will issue an error for the double present declaration of ZWORK arrays.

```
! dyn_sources.f90
...
!$acc data present(ZWORK1,ZWORK2)
...
!$acc kernels
  !$acc loop collapse(3) independent
  DO CONCURRENT (JK=1:JKU,JJ=1:JJU,JI=1:JIU)
    ZWORK1(JI,JJ,JK) = PCURVX(JI,JJ) / XRADIUS
    ZWORK2(JI,JJ,JK) = PCURVY(JI,JJ) / XRADIUS
  ENDDO
!$acc end kernels
...
!$acc end data
...
```

---

over the Tiwi islands. Results are shown here for two grids, $256\times256\times128$ with an horizontal grid spacing of $800\,\text{m}$ and $4096\times4096\times128$ with an horizontal grid spacing of $50\,\text{m}$. It is run during 100 time steps using a time step of $10\,\text{s}$ for the small grid and $4\,\text{s}$ for the large one. For this benchmark, I/O is disabled (except for reading the initial state). Executions start from a situation in which convection has already been initiated and clouds cover a significant portion of the domain. Running times are given without the initialisation phase, unless otherwise stated and are per model time step.

### 3.1 Computer systems

The software versions of essential packages for building and running the Meso-NH executable are listed in Table 2. Computer systems on which the computational performance is measured are detailed in Table 3.

### 3.2 Performance on a single node

All results for a single node are for the $256\times256\times128$ grid. Binding on CPU cores and GPUs is carefully chosen to take into account the fact that Meso-NH is memory-bound (i.e., its runtime is dominated by memory access rather than computation), the





---

**Listing 12** Example of transformed sources after preprocessing the `!$mnh_expand_array` and `present_cr` macros for the Cray compiler. In this case, the second `present` clause is needed for parallelization and to avoid false recurrence detection by the Cray compiler.

---

```
!dyn_sources.f90
...
!$acc data present(ZWORK1,ZWORK2)
...
!$acc kernels present(ZWORK1, ZWORK2)
  !$acc loop collapse(3) independent
  DO JK=1,JKU
    DO JJ=1,JJU
      DO JI=1,JIU
      ZWORK1(JI,JJ,JK) = PCURVX(JI,JJ) / XRADIUS
      ZWORK2(JI,JJ,JK) = PCURVY(JI,JJ) / XRADIUS
      ENDDO
    ENDDO
  ENDDO
!$acc end kernels
...
!$acc end data
...
```

---

**Table 2.** Software used for compiling Meso-NH.

| Software | Jean-Zay | Adastra | Leonardo |
|---|---|---|---|
| Compiler | nvidia-compilers/23.11 | cce/17.0.0 | nvhpc/23.1 |
| MPI communication | openmpi/4.1.5-cuda | cray-mpich/8.1.28 | openmpi/4.1.4 |
| CUDA toolkit or rocm | cuda/12.2 | rocm/5.7.1 | cuda/11.8 |

interconnection between cores on different NUMA (Non Uniform Memory Access) zones, between cores, between CPUs and GPUs and between GPUs and also the way the domain is distributed onto the MPI processes. If several binding configurations have been tested, only the one giving the fastest results is kept. On NVIDIA GPUs, if several MPI processes are started on each one, the NVIDIA Multi-Process Service (MPS) is launched. If this is not the case, performance is severely impacted.

The performance is first detailed on a routine-by-routine basis for a single Adastra node (see its characteristics in Table 3).
The best performance obtained for the 256×256×128 configuration using the FFT solver is shown, that is, using 64 MPI processes for the CPU-only version of Meso-NH and 8 GPUs times 2 MPI processes for the GPU version (Fig. 3).

Overall, Meso-NH is ∼12x faster on GPUs than when using only the CPUs of the same node on Adastra. This reduces the mean time per time step from 8988 ms to 761 ms. This speedup is mainly due to the advection running ∼23x faster on GPUs. The result is a time reduction from 5494 to 241 ms. The second and third biggest time reductions concern cloud microphysics
(from 1358 to 105 ms, ∼13x faster) and turbulence (from 1329 to 119 ms, ∼11x faster). Time is also reduced from 643 to 142 ms for the pressure solver. This ∼4x acceleration is, however, lower than for the previous subroutines. This could be due to the numerous MPI inter-process communications required for global pressure solving. No reduction is achieved for the other





**Table 3.** Main characteristics of the supercomputer nodes used for the porting and performance tests. Note that each AMD MI250X contains 2 GCD (Graphics Compute Die) seen as 2 GPUs by the system.

| Machine | Jean-Zay GPU A100 | Adastra GPU MI250X | Leonardo GPU A100 | Adastra CPU Genoa |
|---|---|---|---|---|
| CPU | 2 x AMD Milan EPYC 7543 32 cores 2.8 GHz | 1 x AMD Trento EPYC 7A53 64 cores 2.0 GHz | 1 x Intel Ice Lake 8358 32 cores 2.6 GHz | 2 x AMD Genoa EPYC 9654 96 cores 2.4 GHz |
| Memory capacity | 512 GiB DDR4-3200 | 256 GiB DDR4-3200 | 512 GiB DDR4-3200 | 768 GiB of DDR5-4800 |
| Memory bandwidth | 409.6 GB/s | 204.8 GB/s | 204.8 GB/s | 900 GB/s |
| GPU | 8 x NVIDIA A100 | 4 x AMD MI250X | 4 x NVIDIA/A100 | |
| Memory capacity | 8 x 80 GiB | 8 x 64 GiB | 4 x 64 GiB | |
| Memory bandwidth | 8 x 1.52 TB/s | 8 x 1.6 TB/s | 4 x 1.52 TB/s | |
| TFLOP/s (64 bit) | 8 x 9.7 | 8 x 23.9 | 4 x 9.7 | |
| Bandwidth CPU/GPU | 8 x 64 GB/s | 4 x 72 GB/s | 4 x 64 GB/s | |
| Bandwidth GPU/GPU | 600 GB/s (with switch) | 100 GB/s to 400 GB/s | 200 GB/s | |
| Inter-node bandwidth | 4 x 25 GB/s | 4 x 50 GB/s | 2 x 25 GB/s | 1 x 50 GB/s |

subroutines. These mainly consist of subroutines not ported to GPU. As the number of processes is 16 for the fastest GPU run versus 64 for the fastest CPU one, and a small fraction of subroutines are ported to GPU, two opposing effects compete and no

gain is expected on this side.

Results on Jean-Zay (see its characteristics in Table 3) show similar performance for the GPU run (553 ms instead of 761 ms per time step (Sankey diagram not shown). As Meso-NH is a memory-bound code, and the memory bandwidths on the NVIDIA A100 and AMD MI250X are relatively similar, this result is expected. The GPU speedup on Jean-Zay compared to the fastest results on the CPUs of the same node is only a factor of 6.2. This difference with an Adastra node (speedup of 11.8) can be

attributed to the fact that the memory bandwidth available to the CPU is higher on Jean-Zay nodes compared to Adastra nodes, leading to a performance increase of around 2 between the two different node types.

GPU performance depends on the number of MPI processes per GPU. Overload performance results are shown using 1, 2, 4 and 8 GPUs of a single node of Adastra and Jean-Zay (Fig. 4). Runtimes for the 256×256×128 configuration using the FFT pressure solver are presented for up to 4 processes per GPU for Adastra and 16 for Jean-Zay. Overloading the Adastra GPU

with more than 4 processes leads to very deteriorated performance and is therefore not shown. On Jean-Zay, due to constraints imposed by the supercomputing center, it is not possible to run jobs with more than 32 processes per node.

The more GPUs, the faster the code. An exception is the similar elapsed time obtained for 4 and 8 GPUs on Jean-Zay, particularly as the number of MPI processes per GPU increases. Two possible explanations should be investigated: the inter-connection between GPUs and/or between CPUs and GPUs is saturated, or the workload by process becomes too low due to

the decreasing size of the MPI subdomains. Increasing the number of MPI processes while maintaining the same number of GPUs generally reduces the total elapsed time. The greatest time reduction is achieved by doubling the number of processes





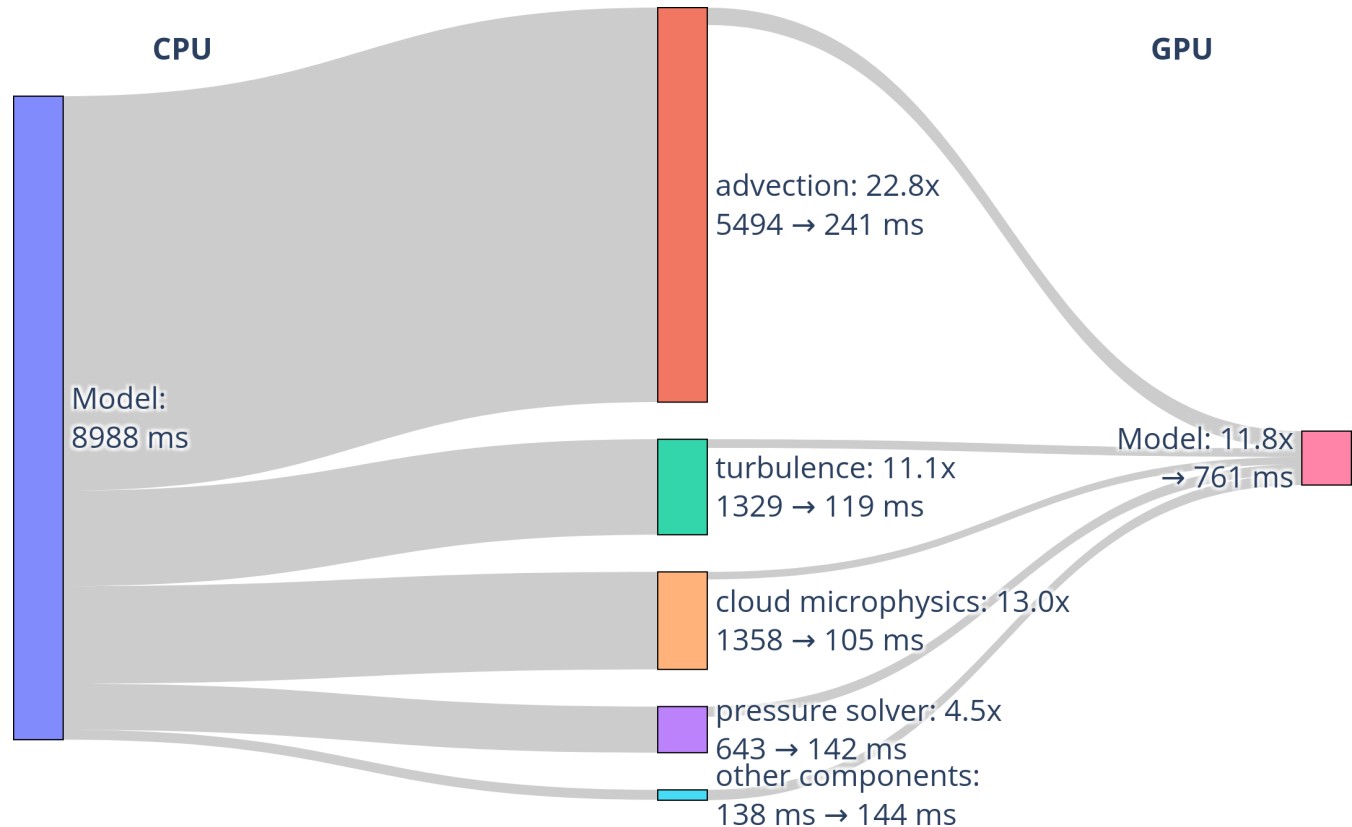

**Figure 3.** Sankey diagram showing the mean runtime per time step achieved on a single Adastra node for the code running on CPU on the left and running on GPU on the right. Results are shown for the 256×256×128 configuration using the FFT pressure solver.

from 1 to 2 for 1, 2 and 4 GPUs. This reduction is most dramatic for the other components, i.e., those parts of the code that are very partially ported to GPU. By increasing the number of processes, the workload of the other components can be distributed over a greater number of CPU cores, reducing the total elapsed time. This reduction with MPI overloading is also found for

the pressure solver, and only on Jean-Zay for cloud and turbulence. Advection is also accelerated when using 2 processes per GPU instead of 1 on Jean-Zay for 1, 2 or 4 GPUs and only for 1 GPU on Adastra. Otherwise, the cost is higher. Using 4 or more processes per GPU does not significantly reduce elapsed time. It even increases it by doubling their number on Jean-Zay, from 8 to 16 for 1 and 2 GPUs, from 4 to 8 for 4 GPUs and from 2 to 4 for 8 GPUs. On Adastra (not shown in Fig. 4), using 8 processes per GPU multiplies running time several times. In summary, this result suggests recommending an overload of 2

MPI processes per GPU when running on a full node.

GPUs of Adastra have 2.5 times the peak computing power of those of Jean-Zay, but their memory bandwidth is very close (see Table 3). As Meso-NH is memory-bound, similar results are expected. For a relatively low number of GPUs, performance without the other components (and therefore with only the GPU part of the code) shows runs on Adastra 2 times slower than on





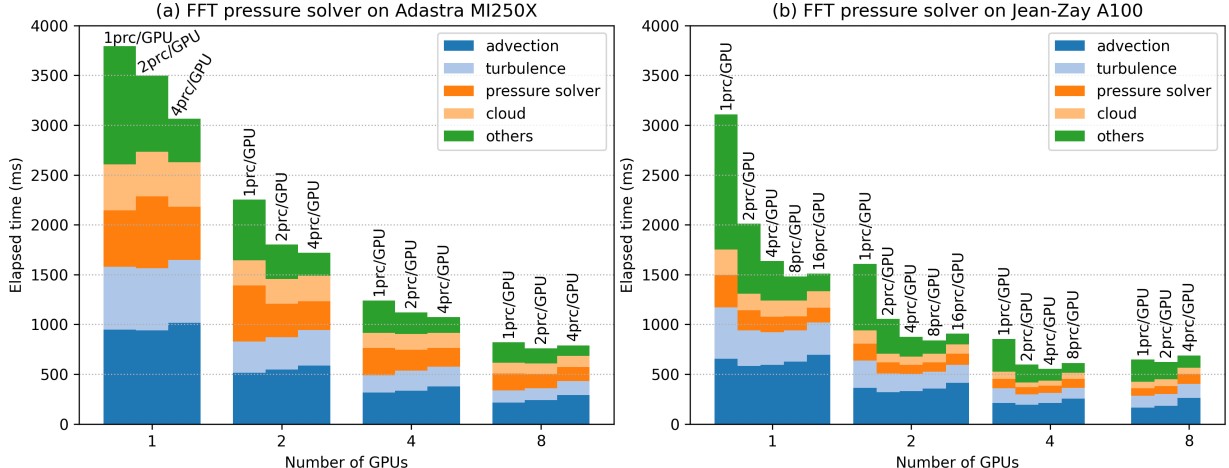

**Figure 4.** Mean runtime per model time step of Meso-NH as function of the number of GPUs for **(a)** Adastra and **(b)** Jean-Zay. The abbreviation "prc" stands for MPI process. Results are shown for the 256×256×128 configuration using the FFT pressure solver.

Jean-Zay. But as the number of processes and GPUs increases, elapsed times on the two systems come closer together. These
differences likely come from the different architectures, GPUs and software environments.

The dependence of GPU performance on the number of MPI processes per GPU is examined in more details for the results of the FFT and MG pressure solvers (Fig. 5). As expected, the results for the FFT pressure solver are similar to those obtained above: (i) the more GPUs, the faster the FFT pressure solver (with the exception of 4 and 8 GPUs on Jean-Zay); (ii) doubling the number of processes from 1 to 2 for 1, 2 and 4 GPUs significantly reduces elapsed time (with the exception of 1 GPU on
Adastra). In other words, overloading GPUs with at least 2 MPI processes makes the FFT pressure solver faster.

Alternatively, overloading GPUs with 2 MPI processes does not greatly affect the speed of the MG pressure solver. Elapsed time increases slightly using up to 4 MPI processes per GPU (with the exception of 8 GPUs on Jean-Zay) and becomes much longer with 8 or 16 MPI processes per GPU (not shown, but also true and much worse with 8 GPUs on Adastra). It is therefore not recommended to overload GPUs with MPI processes for the MG pressure solver, especially as the number of GPUs
increases. The MG pressure solver is faster than the FFT solver when the number of MPI processes is low. The scalability of the MG solver seems lower. As the number of MPI processes increases, the performance of the FFT solver improves faster than the MG one, until it overtakes it. Note that this conclusion is different for the 4096×4096×128 configuration run over a large number of nodes (see Section 3.3). Finally, the performance on 8 GPUs of Jean-Zay is, as seen for the model as a whole, not better than on 4 GPUs.

The energy efficiency of the GPU port is also examined (Fig. 6). The results are shown for the 256×256×128 configuration using the FFT solver on one Adastra node. The energy consumption measurement is returned by the job scheduling software, here the Simple Linux Utility for Resource Management (SLURM), and corresponds to the aggregated node consumption. It does not include network and storage energy consumption. These can be neglected here, as the runs are done on a single





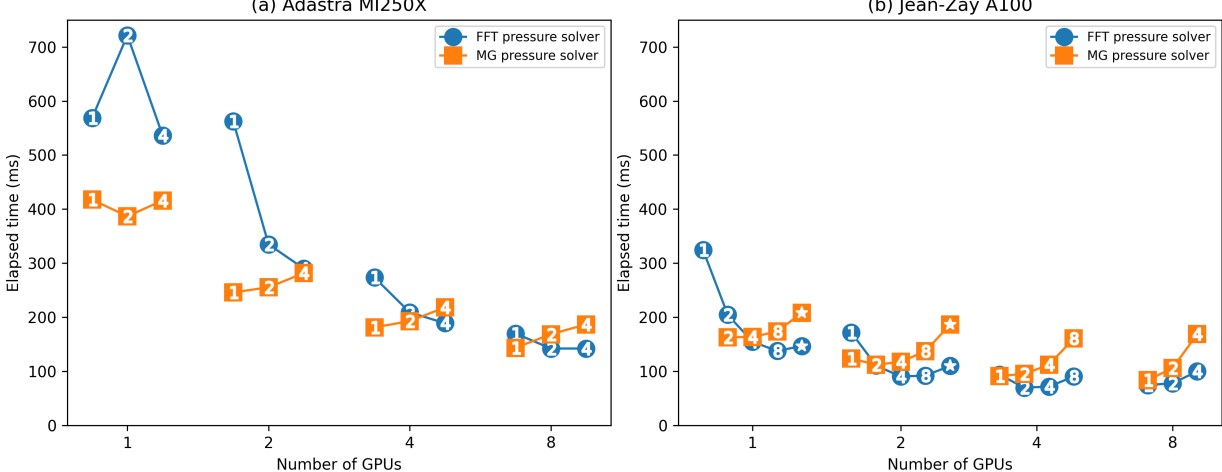

**Figure 5.** Mean runtime per time step of the FFT and MG pressure solvers as a function of the number of GPUs for **(a)** Adastra and **(b)** Jean-Zay. Numbers indicate the numbers of MPI processes per GPU and the star symbol represents 16 MPI processes per GPU. Results are shown for the $256{\times}256{\times}128$ configuration.

computer node and I/O is limited to reading data in the initialization phase. The results compare the CPU and GPU versions
of Meso-NH. They are detailed according to the number of MPI processes. The results for the GPU version use all the 8 GPUs available on the node. Results for the GPU version running with 64 MPI processes are not shown, as overloading AMD MI250X GPUs with 8 MPI processes reduces performance by significantly increasing execution time. Nor are they shown for the CPU version run with 16 MPI processes (or even less), as this configuration partially loads the computing node, leading to almost equivalent power use for double the run time compared with the CPU version running with 32 MPI processes. To obtain
a fair comparison between CPU and GPU energy use, all running on the same node configurations, the CPU measurements have to be corrected to remove GPU energy use in idle mode. This is estimated by taking the power usage reported by the `rocm-smi` command when the GPUs are idle. An average power usage of 90 W is found for each Graphics Compute Die (GCD; a GCD contains 2 GPUs). Note that, unlike the results presented elsewhere, the initialization phase is included in the measurements.

Overall, the GPU version is 3 times more energy-efficient than the CPU-only version. Although GPUs need more power than CPUs and significantly increase the energy requirements of compute nodes, running on GPUs leads to very important gains in terms of energy use. Here, the instantaneous power is around 4 times higher with GPUs, but running time is 12 times shorter. This perfectly illustrates the benefits of using GPUs: shorter running times combined with greater energy efficiency. The number of MPI processes per GPU has an impact on energy consumption. As it increases, so does power draw (power is
energy divided by the running time). However, as seen before, the optimum situation in terms of running time is to put 2 MPI processes per GPU for this test case on Adastra. Here, the fastest GPU run is also the most energy-efficient. If the MG pressure solver is used instead of the FFT solver, the fastest run also corresponds to 2 MPI processes per GPU, but the run with a single





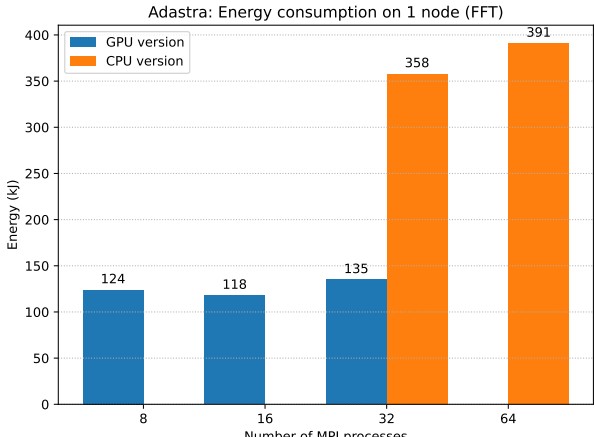

**Figure 6.** Energy use of Meso-NH on a single Adastra node for the CPU and GPU versions. Results are shown for the $256{\times}256{\times}128$ configuration using the FFT pressure solver.

process per GPU is slightly more energy-efficient (needing 110 kJ instead of 116 kJ). For CPU-only runs, using all node cores (64 cores per node) is the fastest, but not necessarily the most energy-efficient. For runs with the FFT pressure solver, energy use increases by 9% compared to a depopulated run with just 32 processes. However, the opposite is found when the MG pressure solver is used (reduction of 11% in energy need).

### 3.3 Scaling

The results of the scaling study on Adastra and Leonardo (see their characteristics in Table 3) are shown for the $4096{\times}4096{\times}128$ configuration (Fig. 7). The x-axis corresponds to the number of nodes used by the model. Four curves are shown for the two computer systems: one using the FFT pressure solver ("OpenACC R8I4 FFT") and three using the MG pressure solver. The managed memory version ("Managed R8I4 MG") enables the system to implicitly manage data transfers between CPUs and GPUs. The "OpenACC R4I4 MG" version is compiled with simple precision for floating point numbers. The "OpenACC R8I4 MG" version corresponds to double precision floating point numbers. In addition, performance on the CPU AMD Genoa partition of Adastra is shown and its run time with 64 nodes is used as a reference for speedup calculation. Table 4 and Table 5 list the data corresponding to the "OpenACC R8I4 MG" and "Managed R8I4 MG" curves, respectively, and detail the speedup routine-by-routine.

The speedup of GPUs compared to CPUs using 64 nodes is between 2.1 and 19.0, depending on the number of nodes and compiled versions of Meso-NH. This demonstrates the benefits of porting. With the same "OpenACC R8I4 MG" version, the code is 5.6x faster on Adastra using 64 nodes, with 9.9x faster advection. Performance is a little lower on Leonardo, with a speedup of 4.4 for the model and 8.7 for advection. This is not completely unexpected, as Leonardo nodes have 4 GPUs instead of 8 for Adastra. The "Managed R8I4 MG" version mainly shows a higher speedup than the "OpenACC R8I4 MG" version,





**Figure 7.** Results for the 4096×4096×128 configuration run on Adastra GPU AMD MI250X, Leonardo GPU NVIDIA A100 and Adastra CPU AMD Genoa. The black dashed line is the reference for the perfect scaling. The speedup is calculated with respect to the elapsed time for the Adastra CPU AMD Genoa partition using 64 nodes.

with the exception of Leonardo with 128 and 256 nodes. Although transfers are better optimized by manual management and have less overhead than managed memory, current developments still incorporate some unnecessary transfers. This explains why "Managed R8I4 MG" tends to be better. The Meso-NH "OpenACC R8I4 FFT" version is slower than the "OpenACC R8I4 MG" version. The difference in speedup is greatest for 256 nodes, by 50% on Leonardo (12.0 versus 8.3). The speedup in pressure solver is indeed 7.0 for MG and 3.2 for FFT. This clearly shows the benefit of introducing a MG pressure solver for GPU porting when the number of GPUs is high.



**Table 4.** Data, in speedup, corresponding to the "OpenACC R8I4 MG" curve in Fig. 7

| Machine | Nodes | Model | Adv. | Turb. | Cloud | Solver |
|---------|-------|-------|------|-------|-------|--------|
| Adastra | 64 | 5.6 | 9.9 | 6.2 | 6.3 | 4.6 |
| | 128 | 8.9 | 16.8 | 10.3 | 11.3 | 6.3 |
| | 256 | 12.6 | 23.8 | 16.5 | 17.3 | 7.0 |
| Leonardo | 32 | 2.4 | 4.8 | 2.1 | 2.5 | 3.7 |
| | 64 | 4.4 | 8.7 | 4.2 | 4.9 | 5.3 |
| | 128 | 7.4 | 14.0 | 8.0 | 9.3 | 6.1 |
| | 256 | 12.0 | 23.6 | 15.0 | 17.7 | 7.1 |

**Table 5.** Data, in speedup, corresponding to the "Managed R8I4 MG" curve in Fig. 7

| Machine | Nodes | Model | Adv. | Turb. | Cloud | Solver |
|---------|-------|-------|------|-------|-------|--------|
| Adastra | 64 | 6.0 | 12.1 | 6.7 | 6.9 | 4.9 |
| | 128 | 9.5 | 18.4 | 11.7 | 12.6 | 5.9 |
| | 256 | 12.9 | 26.1 | 19.4 | 18.6 | 6.4 |
| Leonardo | 32 | 2.6 | 5.4 | 2.9 | 14.0 | 2.6 |
| | 64 | 4.6 | 8.9 | 5.3 | 25.0 | 4.0 |
| | 128 | 7.1 | 14.1 | 9.6 | 27.0 | 4.7 |
| | 256 | 10.0 | 21.7 | 15.1 | 65.1 | 4.9 |

The "OpenACC R4I4 MG" version offers greater acceleration than other versions, for a given number of nodes. This results in the highest speedup with 256 nodes, 17.8 and 19.0 for Leonardo and Adastra, respectively. This is expected because Meso-NH is memory bound. Reducing the size of floating point numbers by a factor of 2 also reduces the amount of data that has to be read from or written to memory by almost 2. This also has the benefit of reducing the memory footprint. The benefit of a reduced precision is higher on Leonardo probably because NVIDIA A100s have twice the computing performance with 32-bit floats than with 64-bit floats unlike AMD MI250Xs which have the same computing power for 32-bit and 64-bit floats. This reduction in precision does not significantly reduce the convergence time of the pressure solver for this test case.

Scaling is almost perfect with CPUs: doubling the number of nodes doubles the speed. The same applies to GPUs, up to 64 nodes only. By using more nodes, the code still runs faster, but at a lower rate than expected, i.e., 13.4 against 16 expected with 128 nodes and 19.0 against 32 expected with 256 nodes with the "OpenACC R4I4 MG" version on Adastra.

The energy gain factor of Meso-NH on Adastra using the GPU AMD MI250X and CPU AMD Genoa partitions is shown for the 4096×4096×128 configuration run (Fig. 8). Four curves are shown for the GPU partition: one using the FFT pressure solver ("OpenACC R8I4 FFT") and three using the MG pressure solver, compiled with managed memory ("Managed R8I4 MG") or





using OpenACC directives with simple or double precision ("OpenACC R4I4 MG" and "OpenACC R8I4 MG", respectively).
The value for the CPU AMD Genoa partition using 64 nodes is taken as reference.

**Figure 8.** Energy gain factor of Meso-NH on Adastra, using the GPU AMD MI250X and CPU AMD Genoa partitions, for the
4096×4096×128 configuration run. The energy gain factor is calculated with respect to the value for the CPU AMD Genoa partition
using 64 nodes.

When using the CPU partition, the energy gain is around 1, regardless of the number of nodes used. In other words, energy
cost with CPUs remains the same. This illustrates the perfect scaling of Meso-NH on CPUs seen above. This contrasts with

energy gain obtained using GPUs by a factor 1.3 to 3.6 compared with the CPU reference, whatever the Meso-NH version
and number of nodes (up to the limit of 256 here). With the OpenACC R8I4 MG version, using GPUs gains energy use to a
factor of 2.3, 2.0 and 1.5 for 64, 128 and 256 nodes, respectively. As expected, the use of simple precision instead of double



precision offers an even greater reduction, with gain factor of 3.6, 3.1 and 2.1 for 64, 128 and 256 nodes, respectively. Gain in energy consumption is increased by up to 0.1 with automatic managed memory from the standard version values. Finally,
using the FFT pressure solver results in higher energy use compared to the version with the MG pressure solver. This results in the lowest gain factor with respect to the CPU reference, that is 1.8 and 1.3 for 128 and 256 nodes, respectively.

# 4  Weather applications

The physical realism and numerical efficiency of the Meso-NH code ported to GPU are exploited here to better understand the mechanisms involved in the formation of small-scale wind gusts in storms. Wind gusts are responsible for major damage,
but remain poorly understood due to their local and intermittent nature ($\sim 1$ s), which is inaccessible to standard atmospheric simulations. As part of a grand challenge on the GPU partition of Adastra, simulations at hectometric resolution were carried out for recent storms leading to extreme wind gusts. This resolution makes it possible to explicitly represent the scale cascade from the core of the storms ($>100$ km) to the deep and shallow convective circulations at the origin of the gusts ($<1$ km). Table 6 summarizes the set-up of these hectometric simulations that all use the same transport schemes and physical parameterizations
as those used in the benchmark.

**Table 6.** Summary of the simulations undertaken for the grand challenge Adastra

| Simulation | $\Delta x$ | Grid size | # grid points | Initial time | Duration | # GPU nodes |
|---|---|---|---|---|---|---|
| Atlantic storm Alex | 100 m | 4096×4096×90 | 1.5 Gpts | 18:00 UTC 01 October 2020 | 6 h | 128 |
| Mediterranean storm | 250 m | 2048×2048×90 | 0.4 Gpts | 04:00 UTC 18 August 2022 | 6 h | 16 |
| Amazonian storm | 200 m | 4096×4096×128 | 2.1 Gpts | 00:00 UTC 18 January 2023 | 24 h | 128 |
| Wave coupled Alex | 200 m | 2048×2048×90 | 0.4 Gpts | 18:00 UTC 01 October 2020 | 6 h | 16 |

## 4.1  Storms across scales and latitudes simulated by Meso-NH

Two weather events representative of the major types of storms that hit Europe are first simulated (Fig. 9): a North Atlantic storm associated with the midlatitude cyclone Alex typical of the winter season (windstorm, top panels), and a Mediterranean storm associated with intense convection and characterized as as derecho, more typical of the summer season (thunderstorm,
middle panels). In both cases, the weather event extends over several hundred km and quickly propagates at a pace of nearly 100 km h$^{-1}$, which requires a large simulation domain to capture the life cycle of several hours (left panels). However, the formation of gusts occurs at scales of a few km at most, which requires high resolution to be accurately represented (right panels). The wind acceleration involves both deep and shallow convections but contrasts between the two storms, as exemplified by the clear spatial separation between strong winds and rain in the Atlantic cyclonic storm (top right panel) and their close
proximity in the Mediterranean convective storm (middle right panel).



To complement the study of midlatitude storms, a third event representative of tropical weather is also simulated: a convective storm that spreads over the Amazonian forest during the recent field campaign CAFE-Brazil (Fig. 9, bottom panels). In this case, the event extends again over several hundred km (the domain is 800-km large) but propagates much more slowly due to the weak easterly ambient wind in the tropics compared to the midlatitudes. Also, the surface wind is much weaker and does not lead to severe gusts or related damages. However, the moderate gusts play a crucial role in continuously triggering new convective cells and ensuring the organization and maintenance of the convective storm as a whole. This upscale effect starts at scales of a few km at most—contrarily to the downscale effect of gust formation in midlatitude storms described above—but also requires high resolution to be accurately represented. While the detailed processes involved are currently investigated in separate studies, the results illustrate that different types of storms are realistically simulated by Meso-NH and benefit from the combination of large domain and high resolution.

## 4.2    Surface winds simulated by Meso-NH coupled to WW3

In order to better represent ocean-atmosphere exchanges, which can be crucial for the formation of wind gusts, Meso-NH can be coupled to a wave and/or an oceanic model thanks to the OASIS3-MCT coupler (Craig et al., 2017; Voldoire et al., 2017; Pianezze et al., 2018). This kind of coupled system has been widely used on the CPU partitions of various supercomputers.

For the first time and thanks to the grand challenge, coupled simulations between Meso-NH and the version 7.02 of the WAVEWATCH III (WW3) spectral wave model (WW3DG, 2019) have been successfully performed on the Adastra GPU partition. The method consists of launching the Meso-NH model on CPUs and GPUs in stand-alone mode, and WW3 on CPUs. OASIS3-MCT version 5.0 is used and exchanges are made using the CPUs of both models. Concerning computational cost, as discussed in Sect. 3, for performance reasons, the Meso-NH model uses only 16 CPUs out of the 64 available per GPU node, but nodes are fully reserved (`#SBATCH  --exclusive` option). The vacant CPUs are therefore used by the WW3 model using the `#SBATCH -m plane` option. Since WW3 runs faster than Meso-NH for the same horizontal resolution, and since parallel exchanges between models via OASIS are very efficient, the computational cost of the coupled simulation is equal to the computational cost of the Meso-NH simulation. No additional cost is found for the coupling component.

Figure 10 shows an example of the benefit of using high resolution and coupled simulation for Atlantic storm Alex (Table 6). The simulated 10 m wind speed and wave height for wind sea from the state-of-the-art 9 km operational ECMWF forecast (Fig. 10a) are compared to the 200 m simulated fields from the coupled Meso-NH/WW3 simulation on Adastra's GPU partition (Fig. 10b). The effect of a high horizontal resolution on the 10 m wind speed is clearly visible, with higher variability and magnitude. Maximum wind speed is 40% higher in the simulation at 200 m than at 9 km. It can also be noticed that a high horizontal resolution has a significant effect on the simulated significant wave height for wind sea, with the spatial extension of the 3 m wave height reduced by tens of km due to a reduced fetch for the 200 m simulation. In that Meso-NH/WW3 coupled simulation, Meso-NH uses 256 processes distributed over 16 nodes with 8 GPUs per node. WW3 uses 16 processes distributed over the same 16 nodes. The 6 simulated hours are realized in 14 elapsed hours. Quantifying the importance of wave-atmosphere feedbacks on the life cycle of the Atlantic storm Alex using this coupled 200 m simulation will be the subject of future studies.





## 5 Conclusions

Porting Meso-NH to GPUs is achieved by including OpenACC directives to the most computationally expensive parts of the code: advection, turbulence, cloud microphysics and pressure solver. This approach allows the same code to be run on CPUs and on hybrid CPU-GPU architectures. Using our own `MPPDB_CHECK` library, the bit reproducibility of Meso-NH has already been ensured on CPUs. This property is extended to GPUs, thus guaranteeing accuracy of the porting and the absence of bugs. A critical point lies in the atmospheric pressure solver, which requires the inversion of an elliptic equation. A multigrid pressure solver is integrated, because the fast Fourier transforms approach used in the original version of the code becomes expensive with a high number of GPUs. Currently, the code runs on different NVIDIA GPU and AMD GPU platforms and scales efficiently up to at least 1,024 GPUs (256 nodes on Adastra and Leonardo). Using the same configuration with 64 nodes on Adastra, Meso-NH is 5.6x faster on GPUs, with 9.9x faster advection, and achieves a 2.3x energy efficiency gain compared to CPUs only.

Porting of other functionalities of the code to GPU is in progress. This includes grid nesting capabilities, the ability of using other grid configurations than those imposed by the multigrid pressure solver, and other components such as the two-moment cloud microphysics scheme. Moreover, the current porting concerns version 5.5 while the latest version of Meso-NH is version 5.7. An update of the latest version with MESONH-V55-OpenACC is therefore necessary. In particular, in version 5.7, the physical parameterizations have been externalized to create PHYEX (PHYsique EXternalisée). This library shared with the operational NWP code AROME of Météo-France aims to provide greater modularity, enable coherent management of developments in physics and facilitate adaptation to different computing architectures. This also opens up possibilities for use in other models. Thus the next version of PHYEX will include the GPU modifications from MESONH-V55-OpenACC as well as domain-specific language development in order to integrate the operational constraints inherent to AROME. Finally, it is expected that the MESONH-V55-OpenACC version will work as such on the new machines equipped with Accelerated Processing Unit (APU), with the advantages of automatic data transfer for memory and I/O (Tandon et al., 2024; Fusco et al., 2024).

First scientific applications focus on the simulation of extreme weather events across scales as part of a grand challenge pilot project on the GPU-based Adastra supercomputer, ranked 3rd in the November 2022 GREEN500 (TOP500.org, 2022a). Three representative storms are simulated: a North Atlantic windstorm associated with a midlatitude cyclone, a Mediterranean convective storm characterized as a derecho, and a mesoscale convective system over the Amazon rainforest. Representation of the North Atlantic storm requires downscaling from the synoptic cyclone scale (>100 km) down to local wind gust formation (<1 km). Inversely, the representation of the Amazon storm requires upscaling from the local triggering of convective cells (<1 km), which organize and maintain the system at the mesoscale (>100 km). Finally, the Mediterranean storm involves both up- and downscaling. We show that Meso-NH successfully represents the cascade of scales for the three representative storms for horizontal grid spacing down to 100 m and grid size up to 4096×4096×128 points (2.1 Gpts). On the Adastra GPU partition and for one of the three storms, coupled simulations between Meso-NH and the WW3 spectral wave model are successfully



carried out using the OASIS3-MCT coupler. It should be noted that the additional cost of the coupling is negligible compared with the cost of Meso-NH, since WW3 and OASIS use the free CPUs of the GPU nodes.

Porting Meso-NH to GPUs opens up new opportunities for simulating extreme weather events across scales. These opportunities are all the greater in a context where artificial intelligence (AI) is experiencing rapid development in meteorology. Meso-NH simulations, on a fine scale, over very large domains and integrating various couplings, constitute a unique source of data for the development of AI emulators. A strong need for giga-LESs (large-eddy simulations on a billion grid points) already exists for variables that are very little measured, such as the vertical speed of the cloud envelope or in storms that the C3IEL
(Cluster for Cloud Evolution, ClImatE and Lightning) and C2OMODO (Convective Core Observations trOugh Microwave Derivatives in the trOpics) satellite projects aim to retrieve (Auguste and Chaboureau, 2022; Brogniez et al., 2022; Dandini et al., 2022). This need is also expressed for near real-time simulations of natural hazards for urgent decision-making (Flatken et al., 2023). Finally, this GPU porting paves the way for future European exascale supercomputers. The upcoming arrival of such machines will enable the creation of tera-LES (over a trillion grid points) and a better understanding of the upscaling and
dowscaling processes occurring during extreme weather events.

*Code and data availability.* Since version 5.1 was released in 2014, Meso-NH has been freely available under the CeCILL-C license agreement. CeCILL is a free software license, explicitly compatible with GNU GPL. The CeCILL-C license agreement grants users the right to modify and re-use the covered software. The Meso-NH version MESONH-v55-OpenACC is available at http://mesonh.aero. obs-mip.fr/gitweb/?p=MNH-git_open_source-lfs.git;a=commit;h=498cd0cb968041038ff6c5b0f2a76d5066c55bfd (last access: 16 Septem-
ber 2024) as well as at https://zenodo.org/doi/10.5281/zenodo.13759713, (Escobar et al., 2024). This repository also contains namelists to run the test cases and python scripts to reproduce the figures of this manuscript. The MNH_Expand_Array preprocessor is available at https://github.com/JuanEscobarMunoz/MNH_Expand_Array (last access: 16 September 2024).

*Author contributions.* JE and PW ported the Meso-NH code and ran the performance tests. JP installed and tested the coupled code. FP, JP, TD and CB designed the Meso-NH simulations on large-scale grid and performed the simulations on Adastra. FP led the grand challenge.
JPC prepared the manuscript with contributions from all co-authors.

*Competing interests.* The authors declare that they have no conflict of interest.

*Acknowledgements.* Computer resources for running Meso-NH were allocated by CALMIP through projects P0121 and P20024 and GENCI through projects GEN1605 and 0111437. This work was supported by the French National Research Agency under grant agreement ANR-21-CE01-0002. The authors thank Naima Alaoui (Eolen), Pascal Vezolle (HPE), Pierre-Eric Bernard (HPE) for their support on porting the



code on Adastra (Contract for Progress CINES and HPE). The authors also thank Didier Gazen (CNRS) for his ever support on the local

computer cluster and Didier Ricard (Météo-France) for providing initialization data for the Meso-NH simulations.



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





**Figure 9.** Illustrations of Meso-NH simulations for (top panels) Atlantic storm Alex on 01 October 2020, (middle panels) a Mediterranean storm on 18 August 2022 and (bottom panels) Amazonian storms on 18 January 2023. Light shadings show the integrated content of hydrometeors, blue shadings show the precipitation rate above 20 mm h$^{-1}$ and red shadings show the 10-m wind speed. Left panels illustrate composites of different times (in UTC) on the whole domain, while right panels illustrate zooms at specific times in the red boxes.



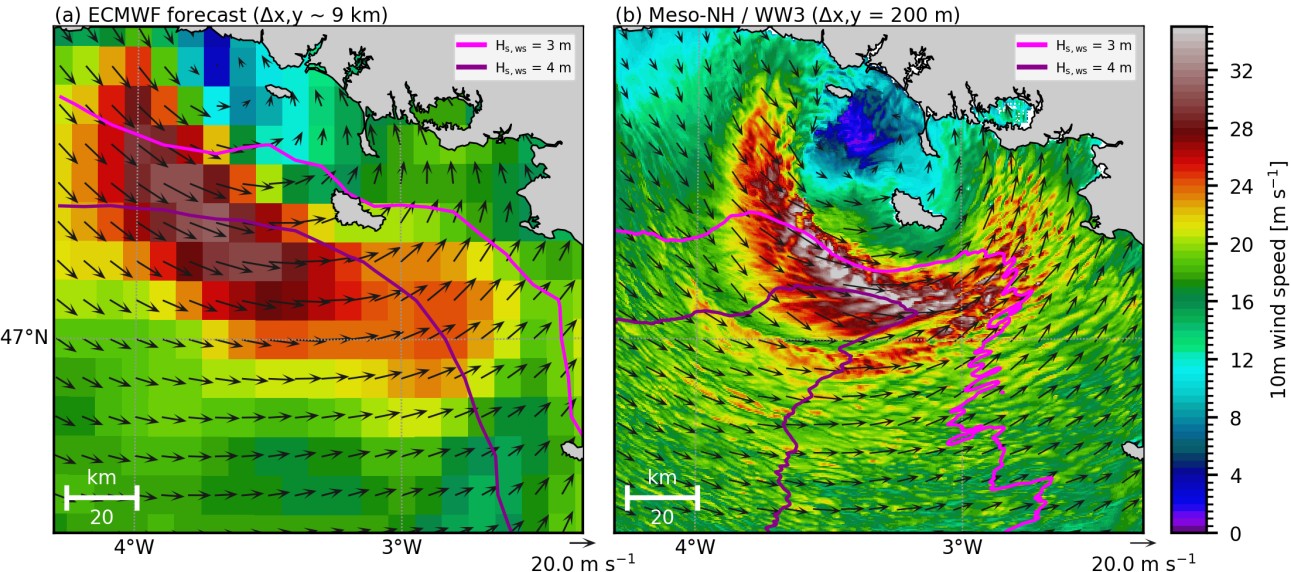

**Figure 10.** Illustrations of coupled simulations of Atlantic storm Alex on 01 October 2020: 10 m wind speed (vectors for direction and color for amplitude) and significant wave height for wind sea (isolines) simulated **(a)** by ECMWF operational forecast and **(b)** by Meso-NH / WW3 on Adastra.