# Peer review of "Porting the Meso-NH atmospheric model on different GPU architectures for the next generation of supercomputers (version MESONH-v55-OpenACC)"

_EGUsphere, 2024_

## Referee Comment (RC1)

**Review of "Porting the Meso-NH atmospheric model on different GPU architectures for the next generation of supercomputers (version MESONH-v55-OpenACC)"**

In this article, the author discuss their successful porting of the Meso-NH model, a mesoscale atmospheric model, from CPU to GPU architecture. The reason is to make use of the faster and more efficient running of certain types of calculations on GPU. The authors indeed report significant improvements in speed and energy efficiency using their new GPU code, and in their article discuss the technical details of the code porting, technical choices that need to be made, bit reproducibility efforts, and at the end a set of demonstrative simulations.

First of all, I would like to congratulate the authors with their new GPU-accelerated code, which no doubt was a big effort. I also like the bit-reproducibility work, which appears crucial in getting reliable results across different hardwares. In reading the article, I came across a few themes that I think need to be clarified, which relate mostly to validation and reproducibility, and a handful of small comments that I list at the end of this review. As my technical knowledge of OpenACC/MPI, compilers, and writing custom libraries is limited I will focus a bit more on the practical side, interpretation, and overal validation.

**Major comments**

**Section 4 shortcomings**
After demonstrating technical side of the porting process and the hardware / performance scaling of the model on various architectures using the author's standard validation case, section 4 is set to demonstrate the "physical realism" and even aims to "better understand the mechanisms involved in the formation of small-scale wind gusts" (lines 433-434). However, the discussion of the simulations is limited to qualitative descriptions which ultimately demonstrate little in terms of new understanding or physical realism. At the very least, I would expect comparison to observations here and quantitative measures of skill, for example compared to what is achievable using the same amount of computing power (in time or energy) on CPU-only simulations (which would demonstrate the benefit). Furthermore, to substantiate the claims of "successful cascade of scales" (e.g. line 10), I would expect a power density spectrum of wind and specific humidity at certain levels.

**Reproducibility**
In light of the previous comment, I thought I'd try and run the code myself on one of the NVIDIA GPU workstations we have available. I use these to run similar GPU-accelerated LES code. After close to an hour, I was not able to compile the library with the documentation supplied in https://zenodo.org/doi/10.5281/zenodo.13759713. Some error code was in French (when you

run `./configure` twice after changing a setting). The top-level README did not guide me through the installation process for a Linux PC, it seemed to be optimized for supercomputer, which is the main purpose, of course. I came across a url for instructions for Linux PCs in the README in the `MESONH-v55-OpenACC` folder, but that url does not work. The compilation seemed almost done, but there were no clear errors at the end - though I suspect an unlinked NetCDF library was the culprit. I don't doubt the compilation process will ultimately be straightforward, but for a user who has never worked with this model before, the instruction for a model as complex as this were too limited given the time I can spend on a review. I don't have access to the supercomputers used by the authors.

**Scaling with radiation**

Given that this model runs mesoscale domains at LES resolution, I would expected that details in physics parameterizations will start to matter. One example is radiative transfer calculations, see:

Maier, R., Jakub, F., Emde, C., Manev, M., Voigt, A., and Mayer, B.: A dynamic approach to three-dimensional radiative transfer in subkilometer-scale numerical weather prediction models: the dynamic TenStream solver v1.0, Geosci. Model Dev., 17, 3357–3383, https://doi.org/10.5194/gmd-17-3357-2024, 2024.

Veerman, M. A., van Stratum, B. J. H., & van Heerwaarden, C. C. (2022). A case study of cumulus convection over land in cloud-resolving simulations with a coupled ray tracer. *Geophysical Research Letters*, 49, e2022GL100808. https://doi.org/10.1029/2022GL100808

Ukkonen, P., & Hogan, R. J. (2024). Twelve times faster yet accurate: A new state-of-the-art in radiation schemes via performance and spectral optimization. *Journal of Advances in Modeling Earth Systems*, 16, e2023MS003932. https://doi.org/10.1029/2023MS003932

Please clarify:

- What scheme do you use and is it GPU-accelerated? Line 98 should be more specific here. I suspect ecRAD.
- How does the radiation scheme affect the scaling performance of CPU vs GPU code? Line 287 says you call it only every 900s.

**Minor comments**

**Readability of the overal manuscript**
As mentioned, I lack the technical know-how of the porting process, and so, feel free to not attribute too much value to this comment. However, if your goal is for the article to be readable to a broader audience, I would advice an approach where the logic and decision making of all steps is written in plain language, with the specific syntax/code not in-line but separate. I understand this may be unavoidable given the topic of the article.

For example, in much of the article, command line options, compiler flags, and run modes are included in parenthesis or in-line in such a way that, for me, the readability of the overal text is challenging and sometimes I lose track of what the purpose of a specific section or paragraph is. In section 4.2, meant to showcase the practical application of the model code, various new technical concepts and run flags are introduced in the first paragraph, and then again at the end of the second paragraph. Lines 106 to 114 may also better be placed in section 2.2? Also the concept of bit-reproducibility can, I think, be explained without the use of inline compiler flags.

Line 58: No code is bug-free, unfortunately. Do you mean that at least single to multi CPU vs GPU will give the same results, and so there are no bugs related to which architecture it runs on?
Line 427: "the use of **single** precision"

---

## Author Comment (AC1)

We thank the Referee for his/her time and his/her constructive comments. We have complied with most of the proposed changes. In the following, the comments made by the Referee appear in black, while our replies are in blue.

In this article, the author discuss their successful porting of the Meso-NH model, a mesoscale atmospheric model, from CPU to GPU architecture. The reason is to make use of the faster and more efficient running of certain types of calculations on GPU. The authors indeed report significant improvements in speed and energy efficiency using their new GPU code, and in their article discuss the technical details of the code porting, technical choices that need to be made, bit reproducibility efforts, and at the end a set of demonstrative simulations.

First of all, I would like to congratulate the authors with their new GPU-accelerated code, which no doubt was a big effort. I also like the bit-reproducibility work, which appears crucial in getting reliable results across different hardwares. In reading the article, I came across a few themes that I think need to be clarified, which relate mostly to validation and reproducibility, and a handful of small comments that I list at the end of this review. As my technical knowledge of OpenACC/MPI, compilers, and writing custom libraries is limited I will focus a bit more on the practical side, interpretation, and overal validation.

**Major comments**

**Section 4 shortcomings**
After demonstrating technical side of the porting process and the hardware / performance scaling of the model on various architectures using the author's standard validation case, section 4 is set to demonstrate the "physical realism" and even aims to "better understand the mechanisms involved in the formation of small-scale wind gusts" (lines 433-434). However, the discussion of the simulations is limited to qualitative descriptions which ultimately demonstrate little in terms of new understanding or physical realism. At the very least, I would expect comparison to observations here and quantitative measures of skill, for example compared to what is achievable using the same amount of computing power (in time or energy) on CPU-only simulations (which would demonstrate the benefit). Furthermore, to substantiate the claims of "successful cascade of scales" (e.g. line 10), I would expect a power density spectrum of wind and specific humidity at certain levels.

We agree that Section 4 does not demonstrate the "physical realism" of the simulations or "the mechanisms involved in the formation of small-scale wind gusts". However, a quantitative comparison with observations and a detailed study of the physical processes are beyond the scope of the paper. As stated in the same section: "While the detailed processes involved are currently investigated in separate studies, the results illustrate that different types of storms are realistically simulated by Meso-NH and benefit from the combination of large domain and high resolution." Thus, we implemented two changes:
(1) We rephrased the first paragraph to make it clear that the purpose is to illustrate possible applications of high-resolution simulations on a large grid, without levering expectations about an assessment of their forecasting skill or a detailed examination of the involved physical processes, which are left for future studies.
(2) We added a new figure to show the benefits of high-resolution simulations (now Fig. 10 in the paper, and the left panel of Fig. 1 below): "Focusing on Atlantic storm Alex prior to landfall over Brittany (Fig. 9, top right panel), Fig. 10 shows the scale cascade as spectrum of kinetic energy in the middle of the boundary layer. The Meso-NH simulation (blue curve) exhibits three distinct ranges: the mesoscale for $\lambda >$10 km, the inertial subrange approaching the theoretical slope of the Kolmogorov spectrum (grey line) for $\lambda <$1 km, and an energy accumulation range in between for $10> \lambda >$1 km. Specifically, the energy accumulation range includes the fine-scale wind structures at the origin of gusts illustrated in Fig. 9. At smaller scales, the drop in energy for $\lambda <$400 m in Fig. 10 indicates that the effective resolution of the simulation reaches $4\Delta$x." In addition, we compare the Meso-NH simulation with the highest-resolution operational model data available: "In contrast, the AROME operational analysis that provides the initial and lateral boundary conditions for the simulation (orange curve) diverges from Meso-NH for $\lambda <$10 km: it captures only the mesoscale and misses the energy accumulation and inertial subrange." We prefer this approach to using the same amount of computing power on CPU-only simulations, as suggested by the Reviewer, which would lead to a large energy consumption for limited added value. Finally, the right panel of Fig. 1 shows the energy spectrum at a lower level: while the

divergence with AROME for $\lambda < 10$ km is similar, the Meso-NH does not clearly follow the slope of the inertial subrange for $\lambda < 1$ km. This is due to the smaller scale of eddies closer to the surface, which would require even higher resolution to be represented explicitly. For the sake of brevity, only the left panel showing the spectra in the middle of the boundary layer is included in the paper.

[Figure]

Figure 1: Energy spectrum for Atlantic storm Alex at 515 m agl (left panel) and 97 m agl (right panel) in the Meso-NH simulation (blue curve) and the AROME operational analysis (orange curve). The grey line shows the theoretical slope of the inertial range, while the vertical lines indicate the approximate wavelengths of the lower limit of the mesoscale ($\lambda \approx 10$ km) and the upper limit of the inertial subrange ($\lambda \approx 1$ km), as well as the effective resolution of the simulation ($\lambda \approx 400$ m).

**Reproducibility**

In light of the previous comment, I thought I'd try and run the code myself on one of the NVIDIA GPU workstations we have available. I use these to run similar GPU-accelerated LES code. After close to an hour, I was not able to compile the library with the documentation supplied in https://zenodo.org/doi/10.5281/zenodo.13759713. Some error code was in French (when you run ./configure twice after changing a setting). The top-level README did not guide me through the installation process for a Linux PC, it seemed to be optimized for supercomputer, which is the main purpose, of course. I came across a url for instructions for Linux PCs in the README in the MESONH-v55-OpenACC folder, but that url does not work. The compilation seemed almost done, but there were no clear errors at the end - though I suspect an unlinked NetCDF library was the culprit. I don't doubt the compilation process will ultimately be straightforward, but for a user who has never worked with this model before, the instruction for a model as complex as this were too limited given the time I can spend on a review. I don't have access to the supercomputers used by the authors. We are sorry to hear that you are having trouble compiling Meso-NH. The problem with the NetCDF library sometimes arises because the compilation of the NetCDF library included in the Meso-NH package may conflict with other NetCDF libraries already compiled on the Linux PC, or because other utility libraries (compression, ...) are missing. It is not possible to take account of each user's specific system environment during the installation process. In the case of compilation issue, the user usually mails his/her problem to Meso-NH support, who will help solving it.

**Scaling with radiation**

Given that this model runs mesoscale domains at LES resolution, I would expected that details in physics parameterizations will start to matter. One example is radiative transfer calculations, see:

Maier, R., Jakub, F., Emde, C., Manev, M., Voigt, A., and Mayer, B.: A dynamic approach to three-dimensional radiative transfer in subkilometer-scale numerical weather prediction models: the dynamic TenStream solver v1.0, Geosci. Model Dev., 17, 3357–3383, https://doi.org/10.5194/gmd-17-3357-2024, 2024.

Veerman, M. A., van Stratum, B. J. H., & van Heerwaarden, C. C. (2022). A case study of cumulus convection over land in cloud-resolving simulations with a coupled ray tracer. Geophysical Research Letters, 49,

e2022GL100808. https://doi.org/10.1029/2022GL100808

Ukkonen, P., & Hogan, R. J. (2024). Twelve times faster yet accurate: A new state-of-the-art in radiation schemes via performance and spectral optimization. Journal of Advances in Modeling Earth Systems, 16, e2023MS003932. https://doi.org/10.1029/2023MS003932

Please clarify:

- What scheme do you use and is it GPU-accelerated? Line 98 should be more specific here. I suspect ecRAD.

- How does the radiation scheme affect the scaling performance of CPU vs GPU code? Line 287 says you call it only every 900s.

We use the radiative code described by Gregory et al. (2000), Revision of convection, radiation and cloud schemes in the ECMWF model, Quart. J. Roy. Meteor. Soc., 126, 1685-1710, https://doi.org/10.1002/qj.49712656607. This reference is now added Line 98. It describes the radiation scheme used at ECMWF before the ecRad scheme was included in the IFS code. As noted in the text, no attempt has been made to port the radiation scheme to the GPU, so it is included in the performance scaling results, along with other components (Line 286, Figure 3 and "Others" in Figure 4).

**Minor comments**

**Readability of the overal manuscript**
As mentioned, I lack the technical know-how of the porting process, and so, feel free to not attribute too much value to this comment. However, if your goal is for the article to be readable to a broader audience, I would advice an approach where the logic and decision making of all steps is written in plain language, with the specific syntax/code not in-line but separate. I understand this may be unavoidable given the topic of the article.
For example, in much of the article, command line options, compiler flags, and run modes are included in parenthesis or in-line in such a way that, for me, the readability of the overal text is challenging and sometimes I lose track of what the purpose of a specific section or paragraph is. In section 4.2, meant to showcase the practical application of the model code, various new technical concepts and run flags are introduced in the first paragraph, and then again at the end of the second paragraph. Lines 106 to 114 may also better be placed in section 2.2? Also the concept of bit-reproducibility can, I think, be explained without the use of inline compiler flags. As you write, we added run flags and other technical elements to show the practical application of porting the Meso-NH code to GPUs. In section 4.2, we have refrained from using these details in the revised version. We prefer to keep lines 106 to 114 in sub-section 2.1, as they are not limited to sub-section 2.2, but include the modifications described in all the following sub-sections. When describing the verification of bit-reproducibility, we prefer to keep the indication of the inline compiler flags, as they are essential to the execution of the MPPDB_CHECK library.

Line 58: No code is bug-free, unfortunately. Do you mean that at least single to multi CPU vs GPU will give the same results, and so there are no bugs related to which architecture it runs on? We agree that no code is bug-free. Here, we state that the implementation for massively parallel executions on CPU or GPU supercomputers is bug free. To clarify, the sentence is now "To our knowledge, Meso-NH is the only atmospheric (or oceanic) model offering bit-level reproducibility. This outstanding capability guarantees that Meso-NH is parallelization bug-free, i.e. there are no bugs in its implementation for parallel executions on CPUs and GPUs."
Line 427: "the use of **single** precision" Changed

---

## Author Comment (AC2)

We thank the Referee for his time and his constructive comments. We have complied with most of the proposed changes. In the following, the comments made by the Referee appear in black, while our replies are in blue.

This work represents a major GPU porting effort of the Meso-NH model. This model is originally written in Fortran with MPI for distributed-memory parallelization. The work ports significant parts of the model for NVIDIA and AMD architectures using OpenACC. In addition to the directives needed to port GPU kernels, a pre-processor was developed, along with a multi-grid pressure solver as an alternative to the FFT-based one. An extensive performance analysis on different systems is provided.

I found the work insightful and the paper well-organized and written. However, some parts lack the detail needed to fully understand the numerical and computational approach. Without clarifying these details, it becomes quite hard to understand some of the choices that were taken in the effort. I will elaborate below point-by-point.

1. L 78-79: "The current pressure solver consists on [of] a conjugate-residual algorithm accelerated by a flat fast Fourier transform (FFT) precondition." This is insufficient to fully understand the numerical approach to solving the pressure equation. Could you provide more (mathematical) background and mention in which directions FFT(s) are being used and the consequences for the grid spacing and boundary conditions along this and the other directions? Moreover, could you illustrate how this equation is solved in parallel (I could not find a clear answer in the cited references either)?

We now provide the reader with specific references while refraining from adding lengthy mathematical explanations. In section 2.1, we specifically implemented three changes:

(1) We now refer to Skamarock et al. (1997), Bernardet (1995) and to the 20 pages of Chapter 9, Part I of the Meso-NH scientific documentation devoted to the pressure problem. It reads "The current pressure solver consists of a conjugate-residual algorithm (Skamarock et al. 1997) accelerated by a flat Fast Fourier transform (FFT) preconditioner following Bernardet (1995). The horizontal part of the operator to invert in the elliptic pressure problem is processed with FFT while its vertical part leads to the classical tridiagonal matrix. For a detailed description, the reader is referred to Chapter 9, Part I of the scientific documentation available on the Meso-NH web site (`http://mesonh.aero.obs-mip.fr`, last access: 16 December 2024)".

(2) We now mention the initial implementation of the FFT solver for parallel computers done by Giraud et al. (1999). We added "The initial parallel implementation of the FFT pressure solver takes into account two other types of partitioning on each horizontal direction, called $x$-slice and $y$-slice. Communication routines have been implemented to move a field between these different decompositions. It is then possible to perform the FFT for each horizontal direction (Giraud et al., 1999)".

(3) We now mention the adaptation of the FFT pressure solver for massively parallel computers. We added "In the case of FFT, moving data from a vertical beam decomposition to $x$- and $y$-slices limits the number of processes to the smallest horizontal dimension. For example, a model on a $512\times512\times128$ grid can only be run with 512 processes. Instead, a 3-dimensional decomposition of the beam was implemented and optimized. For a run using $p_x \times p_y$ processes, the global domain of size $N_x \times N_y \times N_z$ is divided into $z$-pencils of size $(N_x/p_x) \times (N_y/p_y) \times N_z$. The FFT is first performed on each $x$-pencil of size $N_x \times (N_y/p_y) \times (N_z/p_x)$ in the $x$ direction, then on each $y$-pencil of size $(N_x/p_y) \times N_y \times (N_z/p_x)$ in the $y$ direction. Next, the tridiagonal system is solved in the Fourier space for each $z$-pencil. Finally, inverse FFTs are calculated on each $y$-pencil, then on each $x$-pencil. As a result, the above example can now be run with up to $512\times128 = 65536$ processes."

2. L. 80. Can you not simply state that it is written in Modern Fortran? If you want to be pedantic, you'd need to state that it has features from older standards (77, 90), too.

Our intention is not to be pedantic. We simply want to point out that Meso-NH uses more recent features than Fortran 95, some of which are useful for porting to GPUs (i.e., `do concurrent`)

3. L. 122. Just to comment that I found that using 'default(present)' in all OpenACC kernel loops really helps with debugging, as one would get a runtime error whenever something is accessed in a kernel that is not on the device.

The 'default(present)' directive is not really applicable or useful here. As we are porting the code piece by piece to the GPU, not all the data resides on the GPU memory. The code will therefore crash if the data

has been calculated on the CPU and is not yet on the GPU memory. And even if the data is on the GPU memory, an update of the CPU or GPU memory copy is required (`!$acc update host/device ...`). Bit reproducibility ensures that no such errors occur.

4. I found Figure 1 quite hard to understand. Could you improve the captions so that it is clear what we are looking at? Is the left a serial computation, and the right one an MPI decomposed one with 2D pencils?
Following your suggestion, the caption of Fig. 1 is now "Schematic of bit-reproducible verification between primary and replica simulations using the `MPPDB_CHECK` library. On the left, the primary simulation is a computation performed on the entire domain, i.e. without any domain decomposition on CPU. On the right, the replica simulation is a parallel computation performed on CPU or GPU on the domain broken down into $4{\times}4$ pencils."

5. L213. Same spirit as comment 1. "the FFT algorithm requires all-to-all communications between MPI processes (...)" Is the FFT algorithm requiring all-to-all communications, or is it the Poisson solver? It is unclear how the pressure equation is being solved numerically (1D or 2D FFTs? + CR along which direction?), and how that is implemented in a distributed-memory paradigm.
To avoid confusion, we changed "The FFT algorithm [...]" into "The FFT pre-conditioner [...]". See our response to comment 1 regarding the details on the FFT pressure solver added in Sect. 2.1.

6. L 218. "The most promising alternative for solving this type of elliptic equation is the use of a geometric multigrid solver for regular structured grid". This claim needs to be substantiated or reconsidered, as it is not obvious, especially for GPUs: As you coarsen in an MG method, the GPU occupancy is being massively reduced, making it perform extremely poorly on GPU-based systems. So, I would say that geometric multigrid solvers do not pair that well with GPUs.
Our claim is now substantiated by adding "In particular, Müller et al. (2015) ported a C/CUDA version of a geometric multigrid algorithm that scaled up to 16384 GPUs."
We also added a paragraph regarding the cost of FFT-based solvers on supercomputers: "Such a negative impact of all-to-all communications in the FFT pre-conditioner has been seen with Meso-NH running on MIRA, a Blue Gene/Q system at Argonne National Laboratory by showing sub-optimal scalability when using 2 billion threads (Lac et al. 2018; see their Fig. 1). Verma et al. (2023) performed a scaling analysis of their GPU-FFT library for grid sizes of $1024^3$, $2048^3$, and $4096^3$, utilizing up to 512 A100 GPUs. They reported a ratio of communication time to total time of 50% when using 8 GPUs and over 90% when using more than 128 GPUs. Ibeid et al. (2020) showed in exascale projections for grid size of $65536^3$ that the FFT total time is due solely to the FFT communication time, which is dominated by the network access cost."

7. L 235. I see that along one direction the (direct) Thomas algorithm is used, while in other two an iterative (MG) method is used. The linear algebra behind this approach is quite unclear to me, so please provide more mathematical details so a reader can easily follow the method without navigating into the code or other references.
For reasons of readability of the manuscript as a whole (a criticism made by the Reviewer #1), we prefer to refrain from adding lengthy mathematical explanations and we refer the reader to the 20-page documentation of the MG method (Müller 2014).

8. L. 250. A comparison between FFT-based and multigrid is performed, but I am missing a lot of details needed for reproducibility and better understanding. What kind of tolerance is being used in the FFT-based flavor (CR method), and in the MG one? What kind of smoother is being used in the geometric multi-grid method? These details need to be clear for better interpreting the results.
To clarify, we added "It should be noted that the comparison between FFT and MG pressure solvers is presented only in terms of computational performance. No reproducibility of pressure between solvers is expected. Similar accuracy, i.e. the same threshold in the residual divergence of the pressure value, is however demanded by both solvers."

9. L 288. "The test case uses advection, turbulence, cloud microphysics, pressure solver and other components". Consider being more exhaustive here.
To clarify, the sentence is now "The test case uses advection, turbulence, cloud microphysics, pressure solver

(see section 2.1 for more details) and other components. These other components include elements not covered by the above-mentioned processes, such as gravity and Coriolis terms (executed on GPUs), radiation (called every 900 s only, executed on CPUs), time advancement of all variables and I/O operations (which are largely disabled in our simulations)."

10. L 322. I read that there can be several MPI tasks per GPU. It is unclear how this is implemented in practice. A sketch with the domain decomposition colored by MPI tasks, along with the GPUs that handle each group of tasks, would be very insightful.

The binding configuration was explained Line 298 ("Binding on CPU cores and GPUs is carefully chosen [...]. If several binding configurations have been tested, only the one giving the fastest results is kept."). Then we added "For example, the best run on an Adastra node with 16 MPI processes uses a 4×4 subdomain grid. To optimize MPI communications, processes with neighboring subdomains are mapped to nearby GPUs, prioritizing proximity and direct network links. To optimize memory bandwidth, each process is pinned to a separate CPU core, evenly distributed across the 8 L3 caches (2 processes per cache) and 4 NUMA nodes. Finally, MPI processes are paired with their closest GPU for optimal host-device memory transfers."

11. Please re-consider the performance analysis in light of the fact that with MG the GPU occupancy decreases at coarse levels, and if this can explain some of the observations.

It is possible that the low occupancy rate at coarse levels explains the behavior of the MG solver. However, we can note that the occupancy per GPU is fixed when the number of GPUs remains unchanged, but that the solver performance decreases if the number of CPUs is increased. Other phenomena (MPI performance, compiler optimizations, software stack...) could probably explain what is going on and are not easy to differentiate. As a result, we believe it is difficult to determine with any certainty the reasons for what is observed.

12. Finally, in other fluid dynamics domains, direct FFT-based solvers (i.e., FFT factorization along two directions, and Gauss elimination along the last one) show 3x to 100x speed-up compared to multi-grid approaches. While their communication patterns are more complex, their fast performance and good GPU utilization make them quite attractive for GPU-based systems. This goes a bit in contrast with the present observations, though the baseline FFT-based solver is not direct here. I would recommend putting this work in perspective w.r.t. other efforts in the literature that have made similar comparisons.

See our response to comment 6 where several papers are cited on the scalability of FFT-based and MG solvers on exascale systems.

Feel free to contact me directly if something is unclear at P.SimoesCosta@tudelft.nl.

Your comments are very clear. Thanks again for your time.

**References**

Bernardet, P.: The pressure term in the anelastic model: a symmetric solver for an Arakawa C grid in generalized coordinates, Mon. Weather Rev., 123, 2474–2490, https://doi.org/10.1175/1520-0493(1995)123<2474:TPTITA>2.0.CO;2, 1995

Giraud, L., Guivarch, R., and Stein, J.: A Parallel distributed Fast 3D Poisson Solver for MesoNH, in: Euro-Par'99 Parallel Processing. Lecture Notes in Computer Science, vol. 1685, pp. 1431–1434, Springer Berlin Heidelberg, Berlin, Heidelberg, https://doi.org/10.1007/3-540-48311-X_201, 1999

Ibeid, H., Olson, L., and Gropp, W.: FFT, FMM, and multigrid on the road to exascale: Performance challenges and opportunities, J. Parallel. Distrib. Comput., 136, 63–74, https://doi.org/10.1016/j.jpdc.2019.09.014, 2020

Lac, C., Chaboureau, J.-P., Masson, V., Pinty, J.-P., Tulet, P., Escobar, J., Leriche, M., Barthe, C., Aouizerats, B., Augros, C., Aumond, P.,Auguste, F., Bechtold, P., Berthet, S., Bieilli, S., Bosseur, F., Caumont, O., Cohard, J.-M., Colin, J., Couvreux, F., Cuxart, J., Delautier,G., Dauhut, T., Ducrocq, V., Filippi, J.-B., Gazen, D., Geoffroy, O., Gheusi, F., Honnert, R., Lafore, J.-P., Lebeaupin Brossier, C., Libois, Q., Lunet, T., Mari, C., Maric, T., Mascart, P., Mogé, M., Molinié, G., Nuissier, O., Pantillon, F., Peyrillé,

P., Pergaud, J., Perraud, E.,Pianezze, J., Redelsperger, J.-L., Ricard, D., Richard, E., Riette, S., Rodier, Q., Schoetter, R., Seyfried, L., Stein, J., Suhre, K., Taufour,M., Thouron, O., Turner, S., Verrelle, A., Vié, B., Visentin, F., Vionnet, V., and Wautelet, P.: Overview of the Meso-NH model version 5.4and its applications, Geosci. Model Dev., 11, 1929–1969, https://doi.org/10.5194/gmd-11-1929-2018, 2018.

Müller, E. H.: TensorProductMultigrid, p. (last access: 16 December 2024), https://bitbucket.org/em459/tensorproductmultigrid/src/master/,2014

Müller, E. H., Scheichl, R. and Vainikko, E.: Petascale solvers for anisotropic PDEs in atmospheric modelling on GPU clusters, Parallel Computing, 50,53–69, https://doi.org/10.1016/j.parco.2015.10.007, 2015.

Skamarock, W. C., Smolarkiewicz, P. K., and Klemp, J. B.: Preconditioned conjugate-residual solvers for Helmholtz equations in nonhydrostatic models, Mon. Weather Rev., 125, 587–599, https://doi.org/10.1175/1520-0493(1997)125<0587:PCRSFH>2.0.CO;2, 1997

Verma, M., Chatterjee, S., Garg, G., Sharma, B., Arya, N., Kumar, S., Saxena, A., K., M., and Verma, M. K.: Scalable multi-node fast Fouriertransform on GPUs, SN comput. sci., 4, 625, https://doi.org/10.1007/s42979-023-02109-0, 2023.

---

## Referee Report (RR1)

Having gone through the revised manuscript and authors' response, I can see the revision is an overal improvement over the original and the scope or purpose of (some section of) the manuscript has become more clear to me now.

I would still like to comment on some of the responses and changes, as I generally feel the major concerns have only been minimally addressed. Apologies if I did not clearly convey my concerns in the original review.

**Regarding "Section 4 shortcomings"**
I appreciate the rephrasing of the text and the addition of Figure 10. Your goal of revised section 4 is more clear to me now, Figure 10 adds a form of validation compared to theoretical expectations and a comparison to a coarser reference, and it's now clear to me that in-depth quantitative validation and physical understanding is reserved for separate studies.

However, "physical realism" is not substantiated (beyond energy cascade at some level in the boundary layer) and still has to be believed by the reader based on the visualization of a few data fields (figures 9, 11), relative to AROME (figure 10), or relative to ECMWF-IFS (figure 11). Referring to separate future studies for validation is not sufficient. A "Devil's advocate" example: just because Figure 11b has a lot of detail does not mean the operational IFS forecast is worse, it's really a meaningless comparison unless there is a panel 11c showing (buoy?) data of wave height in various locations - for starters.

Ultimately, I'm simply not convinced about the purpose of section 4 in the context of the goal of the study. You have ported the model partially to the GPU and are now ready to numerically run at hectometric resolution. Section 4 successfully shows this, and if that is the goal, then OK. However, section 4 in its current form does not show that the model is ready to run at hectometric resolution in terms of physical realism.

**Regarding "Reproducibility"**
I understand that you can't take each user's specific system into account. I was expecting more user-friendly instructions given the code's complexity, and it is hard to figure out what is going wrong if an error is in French or if a compilation issue does not give any information the cause of error.

My concern was meant in part as honest user feedback (even beyond the scope of just this manuscript) and in part for being clear that I can't fulfil my role as reviewer in terms of checking reproducibility. I suppose I could indeed contact support, but I'm afraid we've then gone beyond what can be expected of me as a reviewer given available time.

**Regarding "Scaling with radiation"**
Radiation was just an example of where model resolution and physical parameterizations can

go out of balance. I think my original comment was not phrased properly, but I believe it is important to address the physics of the model when going to such high resolution, and whether it makes sense to do so.

Even for just the radiation-related questions, the only change I can see is that there is now a reference to a paper that details the radiation model, which I find inadequate:

- Given the description of other model components that are listed/referenced, you could have elaborated more on what kind of radiation you are running and what its limitations are.
- At what spatial resolution do you run radiation? How does it compare to hectometric resolution simulations? How do you generally justify running an older radiation transfer model for a global (then) 15+ km resolution scale model at a 100m resolution?

I think these are important themes in physical modelling and tie back to my concerns regarding "physical realism".

---

## Author Response (AR2)

We thank the Referee for his/her time and his/her constructive comments. We have complied with most of the proposed changes. In the following, the comments made by the Referee appear in black, while our replies are in blue.

Having gone through the revised manuscript and authors' response, I can see the revision is an overal improvement over the original and the scope or purpose of (some section of) the manuscript has become more clear to me now.

I would still like to comment on some of the responses and changes, as I generally feel the major concerns have only been minimally addressed. Apologies if I did not clearly convey my concerns in the original review.

**Regarding "Section 4 shortcomings"**

I appreciate the rephrasing of the text and the addition of Figure 10. Your goal of revised section 4 is more clear to me now, Figure 10 adds a form of validation compared to theoretical expectations and a comparison to a coarser reference, and it's now clear to me that in-depth quantitative validation and physical understanding is reserved for separate studies.

However, "physical realism" is not substantiated (beyond energy cascade at some level in the boundary layer) and still has to be believed by the reader based on the visualization of a few data fields (figures 9, 11), relative to AROME (figure 10), or relative to ECMWF-IFS (figure 11). Referring to separate future studies for validation is not sufficient. A "Devil's advocate" example: just because Figure 11b has a lot of detail does not mean the operational IFS forecast is worse, it's really a meaningless comparison unless there is a panel 11c showing (buoy?) data of wave height in various locations - for starters.

Ultimately, I'm simply not convinced about the purpose of section 4 in the context of the goal of the study. You have ported the model partially to the GPU and are now ready to numerically run at hectometric resolution. Section 4 successfully shows this, and if that is the goal, then OK. However, section 4 in its current form does not show that the model is ready to run at hectometric resolution in terms of physical realism.

We agree that Sect. 4 does not actually demonstrate "physical realism". The goal is in fact (i) to show that the code runs at hectometric resolution on very large grids and (ii) to illustrate possible weather applications. Validation with observations is not only beyond the scope of the article, but for the storms illustrated, it is likely more sensitive to the initial conditions (AROME vs. ECMWF for the example shown in Fig. 11) than to the detailed representation of physical processes.

Accordingly, we made the following changes in Sect. 4:

- Fig. 11 and its description were removed, while Sect. 4.1 and 4.2 were merged and slightly reorganized;

- "The physical realism and numerical efficiency of the Meso-NH code ported to GPU" was changed to "The numerical efficiency of the Meso-NH code ported to GPU";

- "[...] the results illustrate that different types of storms are realistically simulated by Meso-NH and benefit from the combination of large domain and high resolution" was changed to "[...] the results illustrate how the simulation of different types of storms may benefit from the combination of large domain and high resolution."

**Regarding "Reproducibility"**

I understand that you can't take each user's specific system into account. I was expecting more user-friendly instructions given the code's complexity, and it is hard to figure out what is going wrong if an error is in French or if a compilation issue does not give any information the cause of error. My concern was meant in part as honest user feedback (even beyond the scope of just this manuscript) and in part for being clear that I can't fulfil my role as reviewer in terms of checking reproducibility. I suppose I could indeed contact support, but I'm afraid we've then gone beyond what can be expected of me as a reviewer given available time.

We recognize that checking reproducibility is a hard task for a reviewer, if not impossible, since you do not have access to the supercomputers we used.

Regarding the French error code you mentioned in your first review, we have translated these comments written in French into English in the configure file. The updated version of the configure file can be found in a new zenodo repository.

Regarding the compile link issue you mentioned in your first review, it is very challenging for us to give answers without a direct exchange with the Meso-NH support (i.e. Juan Escobar and Philippe Wautelet).

To solve this issue, you will find in the new zenodo repository at `https://zenodo.org/doi/10.5281/zenodo.14872313` a Singularity image of a small test that we hope you will be able to run on one of the NVIDIA GPU workstations available to you.

**Regarding "Scaling with radiation"**

Radiation was just an example of where model resolution and physical parameterizations can go out of balance. I think my original comment was not phrased properly, but I believe it is important to address the physics of the model when going to such high resolution, and whether it makes sense to do so. Even for just the radiation-related questions, the only change I can see is that there is now a reference to a paper that details the radiation model, which I find inadequate:

- Given the description of other model components that are listed/referenced, you could have elaborated more on what kind of radiation you are running and what its limitations are.

- At what spatial resolution do you run radiation? How does it compare to hectometric resolution simulations? How do you generally justify running an older radiation transfer model for a global (then) 15+ km resolution scale model at a 100m resolution?

I think these are important themes in physical modelling and tie back to my concerns regarding "physical realism".

Regarding "physical realism" of our simulations, see above our response to your first comment.

The radiation scheme used in our study is indeed the old radiation transfer model of ECMWF, which is still interfaced with Meso-NH. As this scheme comes from ECMWF and this article is about porting the Meso-NH model to GPUs, no attempt has been made to port it to the GPU. In Sect 2.1, we stated that "the [radiation scheme] originates from the ECMWF and no attempt has been made to port it to the GPU". In Sect. 4.1, the weather applications we show focus on "the broad range of mechanisms involved in the formation of small-scale wind gusts in storms", in which radiation plays no or only a minor role. These two points explain why radiation is outside the scope of our article, and why we refrain from detailing any radiation-related issues you suggest.

We recognize, however, that radiation can be a key element in meteorological applications other than storms. This is particularly the case for fog and shallow clouds (see the two Meso-NH studies by Bergot et al., 2015 and Villefranque et al., 2022, cited in the introduction to the article). In the conclusion, we added "A physical parameterization not included in PHYEX is the ECMWF radiation scheme (neither the version by Gregory et al. 2000, nor ecRad by Hogan and Bozzo, 2018). The most recent scheme, ecRad, is a method for efficiently handling the 3D radiative effects associated with clouds, a property essential for fine resolution. It is currently being ported to GPUs at ECMWF and will be included, when available, in a future version of Meso-NH."

We thank the Referee for his time and his constructive comments. We have complied with most of the proposed changes. In the following, the comments made by the Referee appear in black, while our replies are in blue.

Thank you for addressing my comments. I still feel the need to highlight some minor issues with the replies and revised manuscript:

- In response to my former comment #1, in (3), you highlight a direct Poisson solver, not a conjugate residual algorithm, so the baseline numerical approach that is being used is not very clear to me. Is it an iterative method combined with FFT? Or is it a direct method that solves the Poisson equation in one step? Also, the decomposition you mention seems two-dimensional, not three-dimensional.

As written in the first paragraph of Section 2.1, "The current pressure solver consists on a conjugate-residual algorithm (Skamarock et al., 1997) accelerated by a flat fast Fourier transform (FFT) preconditioner (Bernardet, 1995)." To make the point clearer, the sentence is now "The current pressure solver is an iterative method consisting of a conjugate-residual algorithm (Skamarock et al., 1997) accelerated by a flat fast Fourier transform (FFT) preconditioner (Bernardet, 1995)."

In the third paragraph, we wrote "a 3-dimensional decomposition of the pencil was implemented and optimized." The decomposition is therefore three-dimensional.

- In response to my former comment #7, could you find a better reference explaining this approach? I had to download the repo and compile the latex file to see it.

Sorry, we could not find a better reference. To avoid downloading the repository and recompiling the latex file, we added GeometricMG.pdf, the compiled latex file to the new zenodo repository at `https://zenodo.org/doi/10.5281/zenodo.14872313`.

- In response to my former comment #10. I understand how there can be different binding configurations, but it is still unclear to me how more than one MPI task per GPU is implemented: Is there a domain decomposition within this device, or only several hosts managing one device? I would appreciate it if this could be made clear in the manuscript.

To be more explicit, the link between the domain decomposition on the MPI processes and on their counterpart on the GPUs was added with the following paragraph: "The domain decomposition is performed by dividing the domain into blocks of the closest possible dimensions in both horizontal directions and distributing the subdomains on the different MPI processes (one subdomain per process). Then, each MPI process is associated (bound) with a GPU on which it will offload part of its calculations (those that have been ported to GPU). It is then possible to associate several MPI processes with the same GPU which will then share its resources (in a way quite similar to sharing cores on a CPU)."